# Emergent vulnerability to climate-driven disturbances in European forests

Giovanni Forzieri [1✉], Marco Girardello[1], Guido Ceccherini [1], Jonathan Spinoni[1], Luc Feyen [1], Henrik Hartmann [2], Pieter S. A. Beck[1], Gustau Camps-Valls [3], Gherado Chirici [4], Achille Mauri[5] & Alessandro Cescatti[1]

Forest disturbance regimes are expected to intensify as Earth's climate changes. Quantifying forest vulnerability to disturbances and understanding the underlying mechanisms is crucial to develop mitigation and adaptation strategies. However, observational evidence is largely missing at regional to continental scales. Here, we quantify the vulnerability of European forests to fires, windthrows and insect outbreaks during the period 1979–2018 by integrating machine learning with disturbance data and satellite products. We show that about 33.4 billion tonnes of forest biomass could be seriously affected by these disturbances, with higher relative losses when exposed to windthrows (40%) and fires (34%) compared to insect outbreaks (26%). The spatial pattern in vulnerability is strongly controlled by the interplay between forest characteristics and background climate. Hotspot regions for vulnerability are located at the borders of the climate envelope, in both southern and northern Europe. There is a clear trend in overall forest vulnerability that is driven by a warming-induced reduction in plant defence mechanisms to insect outbreaks, especially at high latitudes.

[1] European Commission, Joint Research Centre, Ispra, Italy. [2] Department of Biogeochemical Processes, Max-Planck Institute for Biogeochemistry, Jena, Germany. [3] Image Processing Laboratory, Universitat de València, Paterna, Spain. [4] Department of Agriculture, Food, Environment and Forestry, University of Florence, Florence, Italy. [5] Faculty of Biological and Environmental Sciences, Organismal and Evolutionary Biology Research Programme, University of Helsinki, Helsinki, Finland. ✉email: giovanni.forzieri@ec.europa.eu

European forests cover more than 2 million km$^2$, corresponding to 33% of the continent's land surface[1]. They provide a set of ecosystem services that contribute to human well-being and climate regulation[2]. Despite the fact that forests are highly resilient ecosystems when confronted with long-term changes in environmental conditions, they are vulnerable to sudden changes because the long life-span of trees limits their ability to rapidly adapt[3,4]. Understanding and quantifying the forest vulnerability to such disturbances and the underlying driving mechanisms is crucial to assess climate impacts and develop effective adaptation strategies. This is particularly urgent in light of the expected changes in climate that could substantially increase the future risks of natural disturbances for European forests[5–8].

Land surface models (LSMs)—the land component of Earth system models used to predict future climate trajectories—have started to incorporate a mechanistic representation of forest disturbances through equations of varying complexity[9]. Nevertheless, current model formulations only partially capture these dynamics due to our incomplete understanding of the underlying ecological processes[10–13]. Previous research on well-studied systems have provided important insights into the complex interactions between forest disturbances and environmental controls[14–16]. However, it is unclear to what extent results of such local-scale analyses can be extrapolated to larger areas. Compilations of reports on past tree mortality events can provide large spatial coverage[5,17–19], but the coarse resolution at which data are usually recorded (e.g., country level) masks the spatial variability and limits the assessment of the environmental controls. Satellite datasets of forest disturbances have become increasingly available and with their high spatial resolution and global consistency can support large-scale comparative efforts[20]. However, while the global mapping of forest disturbances is now feasible[21–23], attributing disturbance agents from remote sensing data remains challenging[24,25]. Recent studies have used satellite retrievals to explore the dependence of tree mortality on environmental controls[19,26,27], yet without attributing the vegetation response to different agents of disturbances. In addition, these studies have typically considered a limited set of drivers[26,28] and have aggregated vulnerability relations at regional level. Such approaches typically adopt "a priori" knowledge to identify the functional relationships that link vulnerability and drivers. Therefore, possible amplification or dampening effects that may emerge at local scale from interactions among multiple factors or compound events[29] cannot be fully disentangled. Advances in the integration of machine learning with the expanding availability of Earth observations is fostering the assessment of ecosystem responses to multiple interacting factors, without assuming any explicit functional relation[30]. However, such approaches have yet to be implemented comprehensively at large scales on multiple types of disturbances.

Here, we investigate the vulnerability of European forests (including Turkey and European Russia) to fires, windthrows, and insect outbreaks over the period 1979–2018. We use random forest (RF) regression as a machine learning method[31] to identify the emergent relationships between vulnerability—expressed by the relative biomass loss following the occurrence of a given disturbance (BL$_{rel}$, response variable)—and a suite of forest, climate, and landscape metrics (predictors) (Methods, Supplementary Fig. 1 and Supplementary Table 1). We retrieve these variables by integrating spatially explicit databases of forest disturbance events with multiple satellite-based and reanalysis products. The RF models—implemented for different plant functional types (PFTs) and disturbances—are applied over the whole of Europe annually between 1979 and 2018. This results in 40-year time series of potential vulnerability that describe the spatio-temporal dynamics of biomass loss should a specific natural disturbance occur.

Factorial simulations are used to isolate the key drivers of the underlying ecological processes. Finally, the vulnerability is integrated over the disturbance types in space and time to detect possible forest hotspots of high susceptibility to natural disturbances. Overall, our analysis sheds light on the vulnerability of European forests to natural disturbances and its ongoing trends in response to changing climate conditions. Results reinforce the importance of accounting for the dynamic nature of vulnerability in order to quantify the present and future impact of natural disturbances on key ecosystem services, such as carbon sequestration. We point out that our vulnerability estimates should not be interpreted as risk levels as defined in the IPCC framework[32,33]. They rather reflect the relative biomass loss conditional on a disturbance occurring, and do not integrate information on the occurrence probability of disturbances nor on the exposure.

## Results and discussion

**Model evaluation.** The RF-based vulnerability models were developed by splitting the observed forest disturbances in two separate samples: 60% of records were used for model calibration, while the remaining 40% was used to validate model performances ("Methods"). Models were calibrated and validated separately for different natural disturbances and for all forests as well as those dominated by individual PFTs.

We found that the best models explain on average 34–49% of the variance in relative biomass loss ($R^2$) across the considered disturbances (Fig. 1), with a root mean squared error (RMSE) ranging between 9 and 11% (corresponding to 12–15% when normalized by the observed range). Models generally tend to overestimate low relative biomass loss events and underestimate those with high relative biomass loss (negative and positive relative errors (REs), respectively). These compensatory effects—when computed over the entire validation set—result in a ~2% overestimation of biomass losses due to windthrows and an underestimation of ~2% and ~10% of those caused by fires and insect outbreaks, respectively (PBIAS). Model performances varied not only across disturbance types but also across PFTs, in particular for insect outbreaks where $R^2$ ranges between 0.28 and 0.53 across diverse PFTs (Fig. 1, inset box and Supplementary Fig. 2).

**Response functions to natural disturbances.** By definition, machine learning methods are not based on the mechanistic representation of the phenomena and therefore cannot provide direct information on the underlying processes influencing the system response to drivers. However, some model-agnostic methods can be applied to gain insights on the outputs of RF models. Here, we used variable importance metrics to quantify and rank how individual environmental factors influence vulnerability (Table 1 and Fig. 2a–c). Furthermore, using partial dependence plots (PDPs) derived from the machine learning algorithm RF (see "Methods") we explored the ecosystem response function (BL$_{rel}$) to natural disturbances across gradients of forest, climate and landscape features (Fig. 2d–e and Supplementary Figs. 3–5).

In accordance with previous studies[34,35], increased biomass, tree density, and tree age, typically associated with great fuel loads, correspond to higher vulnerability to fires (Fig. 2a, d and Supplementary Fig. 3). Plant water stress, indicated by low precipitation (Pcum), high maximum temperature (Tmax), low moisture index (MI), and high fire weather index (FWI) act to further increase vulnerability[36]. On the other hand, landscapes with a low spatial homogeneity and high slopes generally show lower vulnerability likely because of a higher resistance and lower susceptibility to fire spread[37].

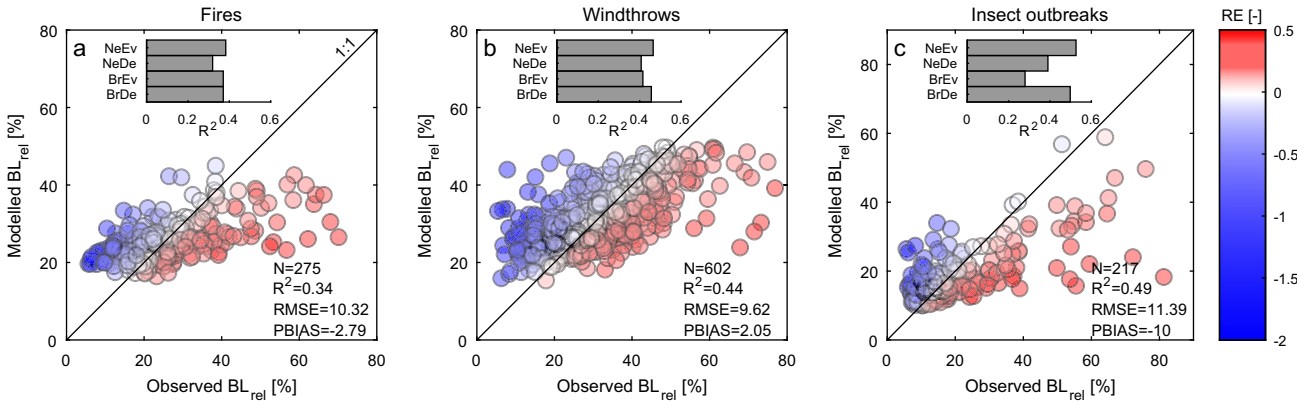

**Fig. 1 Validation of vulnerability models.** Observed versus modelled relative biomass losses (BL$_{rel}$) due to **a** fires, **b** windthrows, and **c** insect outbreaks. Model estimates account for the mixture of different plant functional types (PFTs). Number of binned records (*N*), coefficient of determination ($R^2$), root mean squared error (RMSE) and percent bias (PBIAS) are shown in labels, while relative error (RE) in colour. $R^2$ values in the inset box refer to PFT-specific model performance: broadleaved deciduous (BrDe), broadleaved evergreen (BrEv), needle leaf deciduous (NeDe) and needle leaf evergreen (NeEv).

**Table 1 Environmental variables selected in the vulnerability models.**

| Category | Variable | Temporal resolution |
|---|---|---|
| Forest | Above ground biomass (Biomass) | Yearly |
| | Tree density | Static |
| | Tree age | Static |
| | Leaf Area Index (LAI) | Yearly |
| | Tree height | Static |
| Climate | Fire Weather Index (FWI) | Yearly |
| | Moisture Index (MI) | Yearly |
| | Cumulated precipitation (Pcum) | Yearly |
| | Cumulated snow (Snow) | Yearly |
| | Short-term average anomaly in cumulated precipitation (avg aPcum) | Yearly |
| | Maximum temperature (Tmax) | Yearly |
| | Long-term average temperature (long-term Tavg) | Static |
| | Short-term average anomaly in average temperature (avg aTavg) | Yearly |
| | Maximum wind speed (wind speed) | Yearly |
| | Short-term anomaly in standardized precipitation Evapotranspiration Index (avg SPEI) | Yearly |
| Landscape | Slope | Static |
| | Elevation | Static |
| | Homogeneity | Static |
| | Coefficient of variation (CV) | Static |

Forest, climate and landscape features utilized to characterize the response functions of vulnerability to natural disturbances (see "Methods"). Abbreviations used in text and figures are in parentheses. Details on spatial and temporal aggregation and data sources are listed in Supplementary Methods 1.

Stand biomass is also an important driver of vulnerability to windthrows and shows a positive correlation with biomass loss (Fig. 2b, e and Supplementary Fig. 4). A similar positive relationship emerges for tree age and tree height, which tend to reduce stem flexibility and increase bending moment[38], respectively. As expected, wind speed, a key factor of windthrows, shows a positive relationship with vulnerability. Saturated soils in regions with large amounts of rainfall (Pcum) and increased accumulation of snow tend to weaken root anchorage and cause tree canopy overloading, ultimately favouring tree overturning and stem breakage when exposed to strong wind gusts[39,40].

Colder climates (long-term Tavg) further appear correlated with increased forest vulnerability to windthrows possibly reflecting the shallower rooting systems[41] and the relatively low stem breakage resistance of tree species in these regions[42]. Landscapes with low homogeneity and milder slopes appear less vulnerable to windstorms because they are less prone to domino fall[40] and have typically deeper root depth[43].

Concerning insect outbreaks, contrary to the expectation that forests with high density and LAI may be more affected by insect disturbance[44,45], we found that these forests appear less vulnerable, possibly reflecting good health conditions[4] and limited water stress[46] (Fig. 2c, f and Supplementary Fig. 5). On the other hand, forests with high standing volume (Biomass), typically characterized by older (Tree age) and taller trees (Tree height), are more prone to severe damages. Higher average temperature (avg aTavg) and droughts (avg aPcum and avg SPEI) appear key drivers of forest vulnerability to insect outbreaks possibly because heat and water stress reduce plant resistance to pest damage[47]. At the same time, warm and dry weather anomalies may accelerate insect' development and reduce mortality rate in pest populations. Forests in cold climates (Long-term Tavg), especially those in high-elevation areas, appear particularly subject to such conditions because typically closer to the edge of their thermal tolerance[48]. As also observed for fires and windthrows, forests characterized by high heterogeneity (CV variable) appear less vulnerable to insect outbreaks possibly reflecting the enhanced resistance of mixed-species forests against biotic disturbances[49].

PDPs give the marginal effect of a covariate on the response variable, so the response function (BL$_{rel}$) is only interpretable within and not across covariates. We therefore complemented these analyses and derived the Friedman's *H*-statistics[50] to assess second-order interactions by quantifying how much of the variation of the prediction depends on the two-way interplay ("Methods"). We found that the interaction strength ranges between 13 and 16% depending on the disturbance type (Fig. 3a–c). The type of interacting predictors further modulates the response function. Forest and climate features generally show higher interaction strength in fires and insect outbreaks, while landscape features appear more determinant in generating interacting processes in windthrow events. It is important to note that the observed interactions are positive, amplifying the peaks in the response functions on average by 3–7% (Supplementary Fig. 6a–c). However, for certain combinations of features, the amplification may reach 25% compared to a

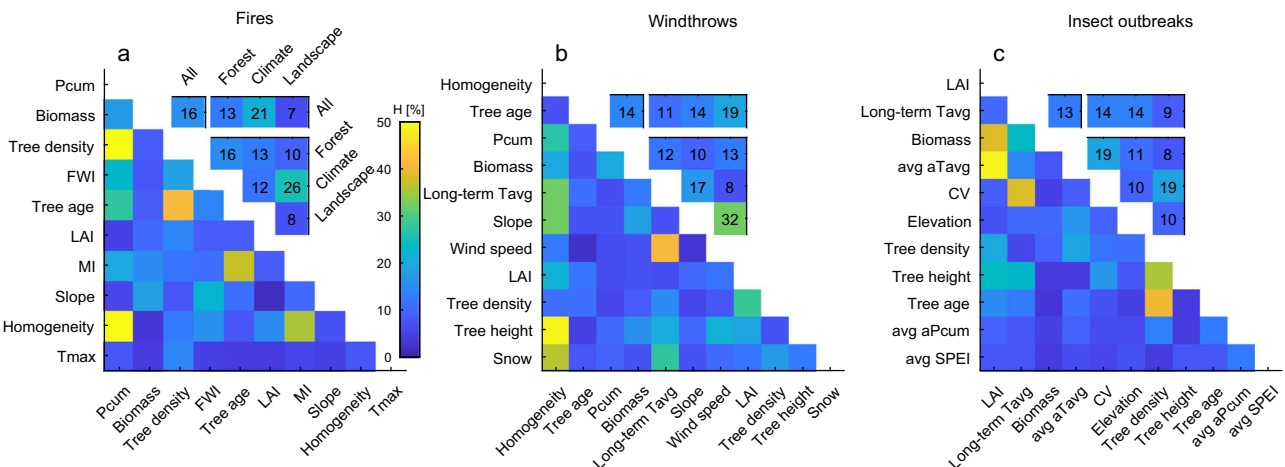

**Fig. 2 Response functions for forest vulnerability to natural disturbances. a** Selected predictors of relative biomass loss and corresponding variable importance based on the random forest regression model of forest vulnerability to fires. Colours distinguish the different categories (forest, climate, landscape) of environmental predictors (see Table 1), while the hatched fill patterns describe the prominent relationship of the response function. **b** and **c** as **a** but for vulnerability to windthrows and insect outbreaks, respectively. **d** Dependence of relative biomass loss (BL_rel) due to fires for the most important predictors in each category—highlighted in yellow outline in panel **a**—as retrieved from zero-centred average partial dependence plots (PDP). Offset values are shown in label for each predictor. **e** and **f** as **d** but for vulnerability to windthrows and insect outbreaks, respectively.

**Fig. 3 Feature interaction strength in the response functions to natural disturbances. a** H-statistic for second-order interactions among environmental predictors of forest vulnerability to fires. Averaged values for different combinations of predictor categories (forest, climate, landscape) and for the whole set of features ("All") are shown in the inset box (reported in colour and numbers). **b** and **c** as **a** but for vulnerability to windthrows and insect outbreaks, respectively. Predictor acronyms are listed in Table 1.

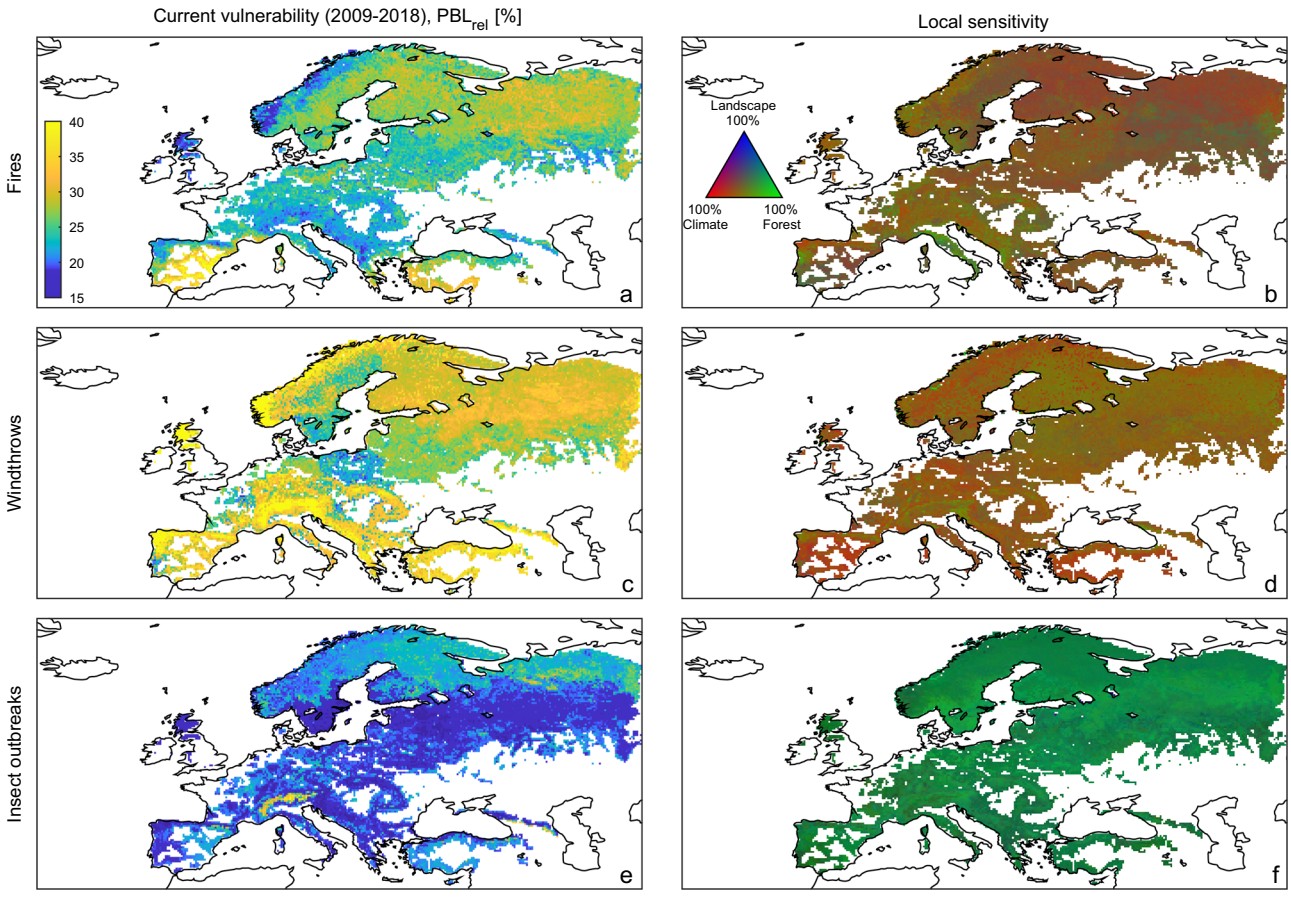

**Fig. 4 Spatial maps of current vulnerability of European forests to natural disturbances and local sensitivity to key drivers. a** Current vulnerability (PBL$_{rel}$) of European forests to fires (averaged over the 2009–2018 period). **b** Marginal contribution of forest, climate and landscape features to the sensitivity of vulnerability to fires. **c**, **d** and **e**, **f** as **a**, **b** but for windthrows and insect outbreaks, respectively. Forests with cover fraction lower than 0.1 are masked in white.

configuration with hypothetical absence of interactions. These findings reinforce the importance of incorporating in a systematic and automated manner the complex interacting processes that are key determinant of the vulnerability of forests to natural disturbances. In addition, the positive sign of the detected interaction terms highlight the key role that climate drivers may have in the future amplification of natural disturbances under global warming scenarios.

**Spatial patterns of vulnerability and its local sensitivity.** When applying RF models in predictive mode and averaging results for Europe over 2009–2018, the vulnerability to windthrows and fires has similar magnitude, with a potential relative biomass loss (PBL$_{rel}$) of 30.2% (29.4–30.3%, 95% confidence interval) and 25.6% (25.0–25.8%), respectively. The vulnerability to insect outbreaks is lower with PBL$_{rel}$ of 19.9% (19.4–20.0%). Considerable spatial variations in PBL$_{rel}$ emerge across regions (Fig. 4a, c, e and Supplementary Table 2). Vulnerability to windthrows is higher in Norway, the North of the British Islands, Portugal, and Southern Europe, and particularly in its mountain zones (e.g., Alps, Caucasus, and Carpathians) (Fig. 4c) where vulnerability may reach 40%. In contrast, forests in southern Sweden and Poland show lower vulnerability, which might be because recent windstorms (e.g., Gudrun in 2004 and Kirill in 2007) have already reduced the biomass of the more susceptible forests. Vulnerability to fires appear higher in Sweden, Finland, European Russia, southern Iberian Peninsula and Turkey with

PBL$_{rel}$ locally exceeding 35%, whereas forests in wet climates such as in central Europe and mountains zones show generally lower values (Fig. 4a). Vulnerability to insect outbreaks generally increases from south-to-north and from low to high-elevation regions, with PBL$_{rel}$ up to 30% (Fig. 4e).

Predicted vulnerability maps show local uncertainty, expressed in terms of standard error (SE) of PBL$_{rel}$, lower than 1% (~4% when normalized with respect to the average) over most of the domain (Supplementary Fig. 7a, d, g). We note that some climate regions characterized by high average PBL$_{rel}$ and high SE are poorly represented in the observational databases of forest disturbances (Supplementary Fig. 7). For instance, cold–wet and warm–dry zones are largely missing in the windthrows dataset and we have few fire records from cold–dry zones. Although we cannot fully evaluate model performances outside the range of the training/testing sets, we stress that dedicated checks were performed on the PDPs at the boundaries of the observational ranges to reduce potential extrapolation errors ("Methods"). Furthermore, spatial statistics of vulnerability based solely on areas with climatological conditions analogous to those of the observational datasets (Methods and Supplementary Fig. 8) showed marginal variations (<1 percentage point when averaged at the Europe level) compared to the estimates derived from the entire spatial domain (Supplementary Table 2). These results corroborate the robustness of our findings. In light of the above-mentioned considerations, we speculate that forests that were predicted to be highly vulnerable by our models but have experienced no or few natural disturbances over the observational

period may now be less adapted to disturbances compared to forests more prone to ecological stresses. Recent modelling studies found that, over centuries, increasing disturbance frequency fosters the reorganization of ecosystems and catalyses the adaptation of forest composition to climate change[51]. While such studies seemingly confirm our interpretation, additional research is needed to confirm whether this mechanism operates at short timescales too.

Vulnerability shows variations in sensitivity to environmental conditions (Fig. 4b, d, f and Supplementary Figs. 9–11). Local sensitivities, here quantified by the first-order derivative of the forest response function ("Methods"), express the degree to which vulnerability changes along the gradient of a given environmental predictor. Forest structure characteristics play a prominent role in influencing such sensitivities, particularly for insect outbreaks, for which the average marginal contribution ($Z_{marg}$) at European level is larger than 60% (Fig. 4f). Biomass, LAI, tree age and tree density are key determinants of such emerging patterns; however, the magnitude of their effects on local sensitivity vary greatly among natural disturbances and geographical regions (Supplementary Figs. 9–11). Climate features are also important in controlling the vulnerability to fires and windthrows with a $Z_{marg}$ greater than 49% and 54%, respectively (Fig. 4b, d), while only marginally affecting the vulnerability to insect outbreaks compared to other environmental factors (8%, Fig. 4f). We point out that such local sensitivity patterns refer to the mechanisms characterizing the vulnerability and not to the triggering processes, which on the contrary have been shown to be largely dependent on climate conditions[52,53]. Residual effects on local sensitivity of vulnerability are explained by landscape metrics. In the case of insect outbreaks, for which landscape metrics have a 31% marginal contribution, elevation plays an important role (Supplementary Fig. 11) because it is associated with the conditions that control insect populations and their rate of spread[54] (e.g., Mountain Pine Beetle).

Overall, results show a current substantial predisposition of European forests to be adversely affected by natural disturbances. They also emphasize the importance of forest structural characteristics in determining forest vulnerability, yet the magnitude of their effects are strictly linked to the local environmental conditions.

**Trends in vulnerability in response to ongoing climate change**. Assessing the temporal evolution of vulnerability is a prerequisite to understand the forest ecosystem response to ongoing climate change. To this aim, we explored the temporal evolution of forest vulnerability in response to changing climate conditions over the period 1979–2018. We found that at European level there is no substantial trend in forests' vulnerability to fires and windthrows ($-4.9 \times 10^{-3}$ and $-1.4 \times 10^{-3}$% year$^{-1}$, respectively) and its dynamics appear dominated by the large interannual variability in climate (Fig. 5a, c). The time series analysis performed at grid-cell level confirms these results, showing mostly non-significant trends ($\delta PBL_{rel}$) and contrasting patterns across Europe (Fig. 5b, d). Locally significant positive trends in vulnerability to fires appear in Iberian Peninsula, Italy, southern France and parts of Belarus and Ukraine, and are associated with an increase in water stress (Fig. 5b and Supplementary Fig. 12a–d). Opposite trends occur mostly in Greece, Turkey, eastern Europe, northern Europe and European Russia. Positive trends in vulnerability to windthrows are evident in the Balkan countries and parts of Portugal and Norway (Fig. 5d), following increases in precipitation, snowfall and wind speed (Supplementary Fig. 12e–g), whereas opposite trends appear mostly in central Europe and inland territories of Norway. However, the significance of these changes remains largely elusive.

In contrast to fires and windthrows, the vulnerability to insect outbreaks at European level grew substantially ($8.8 \times 10^{-2}$% year$^{-1}$) in response to changing climate conditions, particularly from 2000 onwards (Fig. 5e). Most of Europe shows statistically significant increasing trends in vulnerability to insect outbreaks with local $\delta PBL_{rel}$ exceeding 0.2% year$^{-1}$ in north-eastern Fennoscandia and northern European Russia (Fig. 5f). Such a rise in vulnerability appears largely driven by the increase in temperature, which represents the dominant factor in 91% of the area (Fig. 5f and Supplementary Fig. 12h–j). The widespread and significant rise in temperature (Supplementary Fig. 12h) and the monotonic increase of the response function of vulnerability to temperature (Supplementary Fig. 5d) largely explain such trends despite an overall low local sensitivity of vulnerability to climate drivers (Fig. 4f). In particular, the temperature anomaly of +0.5 °C with respect to the 1970–1990 climatology reached around the year 2000 (Fig. 5e, inset box) corresponds to a temperature threshold after which vulnerability started increasing steadily (Fig. 5e). This suggests that, around the year 2000 temperature reached a tipping point that substantially altered forest resilience to pest outbreaks. Indeed, further increases in temperature since then have likely reduced plant defence mechanisms making European forests progressively more vulnerable to insect outbreaks. Even when water availability was not a limiting factor (Supplementary Fig. 12h–j), rising temperatures may have affected plant water status by increasing the vapour pressure deficit and decreasing stomatal conductance[55], which ultimately decreased labile carbon storage, secondary metabolism and plant resistance[56]. This seems confirmed by the documented recent rise in infestations of bark beetles responsible for massive and destructive attacks on coniferous forests of many northern and eastern European regions[57].

**Overall vulnerability to multiple disturbances**. When multiple disturbances are combined into a single composite index, hereafter called overall vulnerability index (OVI), average results show that over 2009–2018 European forests have a vulnerability of about 58% (57.0–58.4%) of their biomass (Fig. 6a). This corresponds to 33.4 billion tonnes of biomass that could potentially be lost due to natural disturbances, mostly located in broadleaved deciduous (BrDe) and needle leaf evergreen (NeEv) forests. The potential biomass loss is driven primarily by the vulnerability of forests to windthrows (40%), followed by fires (34%) and insect outbreaks (26%) (Fig. 6a, inset pie chart). The OVI for Europe shows an increase of $4.2 \times 10^{-2}$% year$^{-1}$ ($4.1 \times 10^{-2}$–$4.3 \times 10^{-2}$% year$^{-1}$) over the observational period (Fig. 6b), a trend dominated by the temporal patterns of vulnerability to insect outbreaks (Fig. 6b, inset pie chart).

By integrating spatial and temporal patterns of OVI, forests in cold climates of Finland, northern European Russia and the Alps, and to some extent warm–dry forests in the interior of the Iberian Peninsula, emerge as particularly fragile ecosystems. They are characterized by a high overall vulnerability and a concomitant progressive intensification due to changes in climate (Fig. 6d, e). The underlying correlation between spatial and temporal patterns seems controlled predominantly by the vulnerability to insect outbreaks ($R^2 = 0.55$, Fig. 6c). The vulnerability in these hotspots is high due to a combination of current environmental conditions and the intensified warming that occurred in the last decades. Hence, global warming poses a serious threat to forest ecosystems with potential critical consequences for climate mitigation actions and local economies that are highly dependent on the forest sector[1,58].

We stress that our estimates cannot account for interactions among multiple disturbances ("Methods") due to the structural limitations of the datasets used to train the models. Cascading and amplifying effects originating from natural disturbance

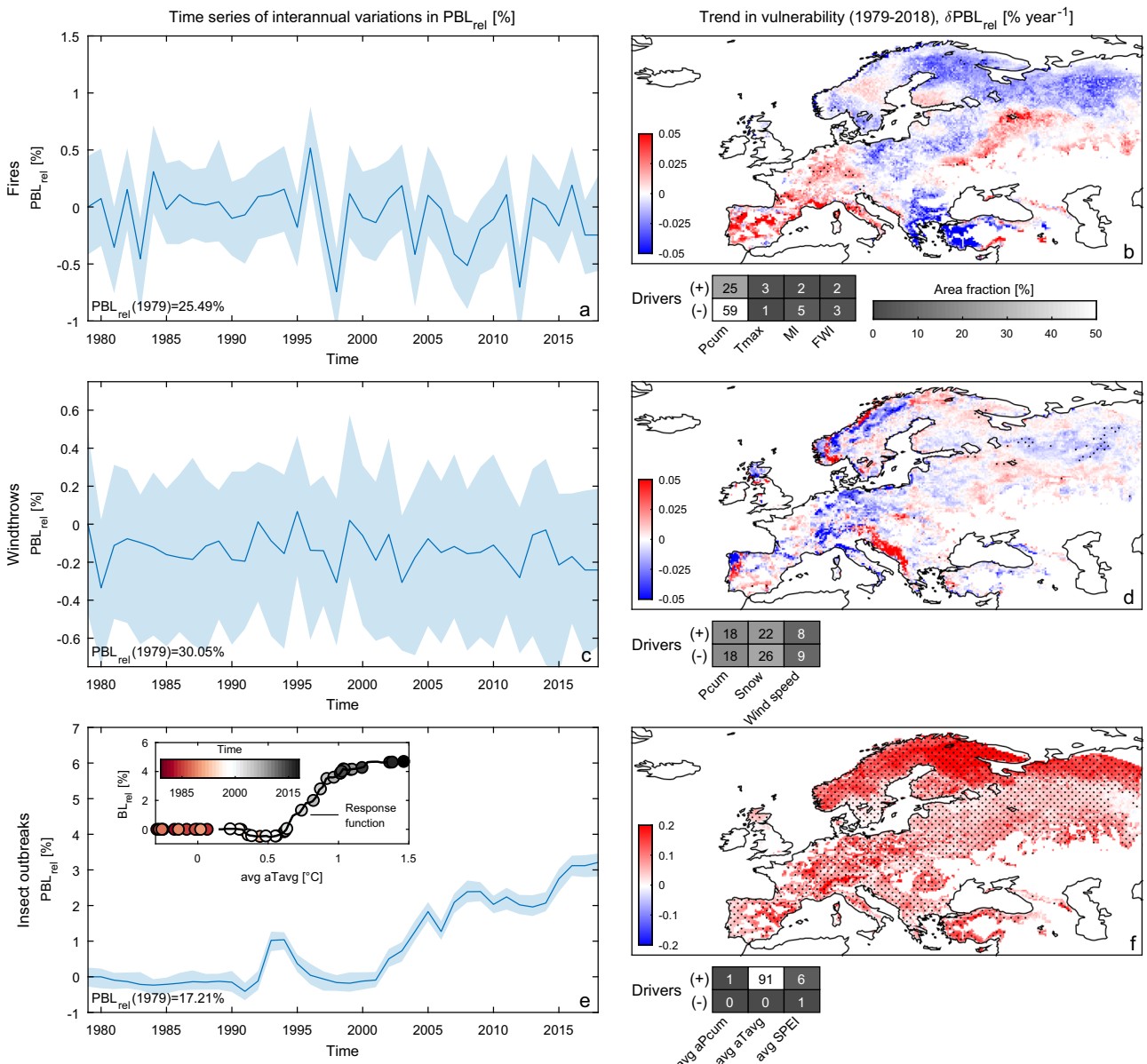

**Fig. 5 Temporal variations in forest vulnerability to natural disturbances over the 1979–2018 period. a** Time series of vulnerability to fires (PBL$_{rel}$) aggregated at Europe level and rescaled to the first year (1979) vulnerability value (shown at the bottom left). The blue line and the shaded patterns reflect the annual mean value and its 95% confidence interval, respectively. **b** Spatial map of the temporal trends in vulnerability to fires ($\delta$PBL$_{rel}$); black dots show pixels where trends are significant (two-sided Mann–Kendall test; $p$ value < 0.05). Corresponding temporal drivers visualized in terms of area fraction (reported in colour and numbers) where the given driver is dominant. The positive and negative effect of each driver on vulnerability is distinguished by the symbols "+" and "−", respectively. **c**, **d** and **e**, **f** as **a**, **b** but for windthrows and insect outbreaks, respectively. Inset box in panel **e** shows the average response function of the vulnerability to insect outbreaks along the observed gradient of temperature anomalies. Annual values of temperature anomalies aggregated at Europe level are overlaid on the response function and visualized in colour to capture their temporal evolution. Predictor acronyms are listed in Table 1.

interacting have the potential to increase the vulnerability of forest ecosystems[59,60] and lead to irreversible shifts in ecosystem states[61]. Therefore, our results are likely to reflect only partially the overall vulnerability resulting from the interactions of fires, windthrows, and insect outbreaks.

**Conclusions and implications**. This study assesses the spatial and temporal dynamics of the vulnerability of European forests to fires, windthrows, and insect outbreaks and disentangles its key drivers. We show that forest structural, physiological and mechanical properties largely control forest vulnerability to these

natural disturbances. These results emphasize the potential of forest management to increase the resilience and long-term stability of European forests and related ecosystem services. Our analysis also shows that ongoing climate change has already affected forest vulnerability and that the positive interplay between climate and other environmental drivers has further amplified forest vulnerability. In particular, rising temperature after 2000 has increased the vulnerability of European forests to insect outbreaks. We hypothesize that a tipping point was reached in that year and further temperature rises weakened plant defence mechanisms to insect outbreaks. Given the expected continuation of warming and the probable intensification of natural

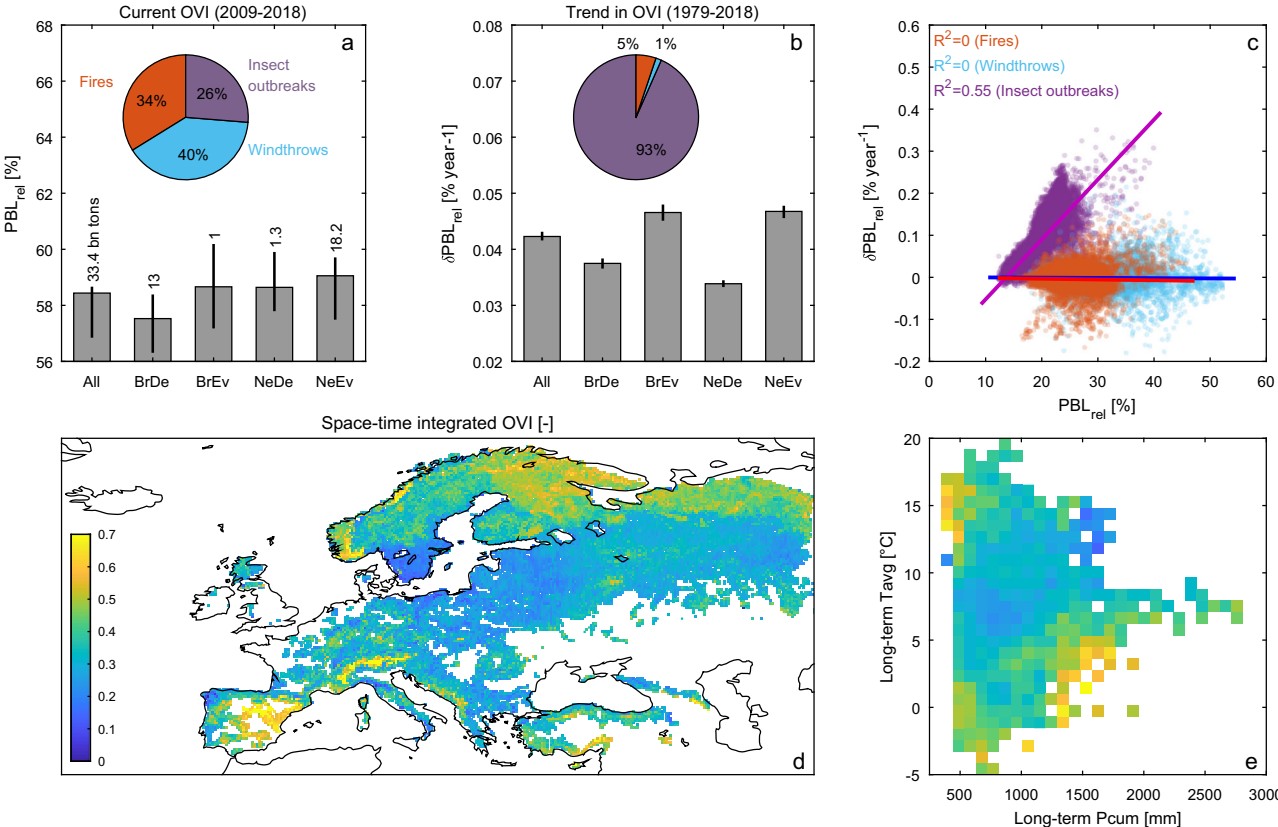

**Fig. 6 Spatial and temporal patterns of the overall vulnerability of forests to multiple natural disturbances. a** Current overall vulnerability index (OVI expressed in terms of $PBL_{rel}$) to multiple disturbances (averaged over the 2009–2018 period) and averaged over the whole domain (All) and separately for different plant functional types: broadleaved deciduous (BrDe), broadleaved evergreen (BrEv), needle leaf deciduous (NeDe) and needle leaf evergreen (NeEv). Bars represent the average value of the 15,797 0.25° grid cells weighted by their forest area extent; whiskers reflect the corresponding 95% confidence intervals while vertical labels report the total vulnerable biomass. The inset chart shows the marginal contribution of each natural disturbance to the OVI computed over the entire domain. **b** as **a** but for the trends in OVI computed for the 1979–2018 period (expressed in terms of $\delta PBL_{rel}$). **c** Current grid-cell vulnerability and their trend for each disturbance type. Labels report the coefficient of determination of the linear regression shown as coloured lines. **d** Spatial map of the space-time integrated OVI. Forests with cover fraction lower than 0.1 are masked in white. **e** Space-time integrated OVI binned as a function of the long-term cumulated precipitation (on the x-axis) and average temperature (on the y-axis).

disturbances in coming decades[3,5], key ecosystem services such as carbon sequestration and biodiversity conservation, could be seriously compromised in the near future. This is particularly the case in northern Europe, which we find to be particularly susceptible and exposed to accelerated warming.

In addition, our findings may serve as a benchmark for LSMs to improve their capacity to represent natural disturbances and ultimately enhance the reliability of future land–climate predictions. All together, these results may help the development of more integrated and effective mitigation and adaptation strategies by informing climate policies on the current vulnerability of the forest carbon stock and on potential ways to increase forest resilience to climate hazards.

## Methods

**Observed forest disturbances.** We focused on the vulnerability of European forests to three major natural disturbances: forest fires, windthrows and insect outbreaks (bark beetles, defoliators and sucking insects). In order to identify/ calibrate/validate vulnerability models (details on model development in the following sections) we used a large number of records of forest disturbances collected over the 2000–2017 period (Supplementary Fig. 1, step1). Fires were retrieved from the European Forest Fire Information System (EFFIS, https://effis.jrc.ec.europa.eu/) and count 15,818 records. Windthrows were acquired from the European Forest Windthrow dataset[62] (FORWIND, https://doi.org/10.6084/m9.figshare.9555008) with 89,743 records. Insect outbreaks were retrieved from the National Insect and Disease Survey (IDS, http://foresthealth.fs.usda.gov) database of the United States Department of Agriculture (USDA) which includes 50,777 records. Each

disturbance record is represented by a vector feature describing the spatial delineation of the damaged forest patch obtained by visual photointerpretation of aerial and satellite imagery or field surveys.

Even if the study focuses on Europe, for insect diseases we used the IDS-USDA database due to the lack of an analogous monitoring system and related dataset for Europe. Therefore, the models of vulnerability to insect outbreaks were identified/ calibrated/validated on US data and then applied in predictive mode to Europe (see following sections for details). To assure the transferability of such models, we developed models for functional groups instead of working on species-specific models. For this purpose, we classified records based on functional groups of the pest (bark beetles, defoliators and sucking insects) and on the PFT of the host tree species. Records were considered if the host plant belonged to the following PFTs: broadleaved deciduous, broadleaved evergreen, needle leaf deciduous and needle leaf evergreen.

**Reconstruction of annual biomass time series.** In order to evaluate the biomass loss expected given a disturbance event occurs, multi-temporal information of biomass is required. However, there is still no single technology for direct and continuous monitoring of such variable in time. In order to reconstruct the temporal variations in biomass over the 2000–2017 period we integrated a static 100-m above ground biomass map acquired for the year 2010 from multiple Earth Observation systems[63] with forest cover changes derived from the Global Forest Change (GFC) maps recorded at 30-m spatial resolution from Landsat imagery[21]. The GFC maps include three major layers: "2000 Tree Cover", "Forest Cover Loss" and "Forest Cover Gain". "2000 Tree Cover" ($TC_{2000}$) is a global map of tree canopy cover (expressed in percentage) for the year 2000. "Forest Cover Loss" is defined as the complete removal of tree-cover canopy at the Landsat pixel scale (natural or human-driven) and is reported annually. "Forest Cover Gain" reflects a non-forest to forest change and refers to the period 2000–2012 as unique feature without reporting the timing of the gain.

The data integration approach built a on the assumption that changes in biomass are fully conditioned by the changes in tree cover. First, we quantified the percentage of tree cover in 2010 ($TC_{2010}$) by masking out all pixels where forest loss occurred over the 2000–2010 period from the $TC_{2000}$ map.

Then, in order to characterize to what extent an increase or decrease in tree cover may affect biomass, we quantified the density of biomass per percentage of tree cover lost ($\rho_{loss}$) and gained ($\rho_{gain}$) as follows:

$$\rho_{loss} = \frac{B_{2010}}{TC_{2010,loss}}, \tag{1}$$

$$\rho_{gain} = \frac{B_{2010}}{TC_{2010,gain}}, \tag{2}$$

where $B_{2010}$ is the static biomass map available for the year 2010 (ref. [63]). $TC_{2010,loss}$ is the $TC_{2010}$ masked over the pixels where there has been a forest loss during the 2011–2017 period. This filtering provides a picture of forests that were intact in 2010 but removed since then. Similarly, $TC_{2010,gain}$ is the $TC_{2010}$ masked over the pixels where there has been a forest gain and identifies the reforested and afforested areas. Since the map of forest gain is a binary map referring to the year 2012, forest gain pixels lack any information on their tree cover as their value in 2000 is zero. We therefore associated to forest gain pixels the maximum of tree cover percentage computed in a moving window with a radius of 2.5 km. This value represents the maximum potential tree cover in the local environmental conditions and refers to the whole 2000–2012 period ($TC_{2012,gain}$). Then, we assumed that forest gain proceeds at a constant rate over time and that the associated tree cover thus grows linearly:

$$\frac{TC_{2010,gain}}{(2010-2000)} = \frac{TC_{2012,gain}}{(2012-2000)} \rightarrow TC_{2010,gain} = 0.83 \cdot TC_{2012,gain}, \tag{3}$$

Both $TC_{2010,loss}$ and $TC_{2010,gain}$ were resampled to the $B_{2010}$ spatial resolution (100 m). Supplementary Figure 13 shows the frequency distribution of $\rho_{loss}$ and $\rho_{gain}$ over a test area in Southern Finland. As expected, the density of biomass associated with forest losses is higher than that associated to forest gain. Indeed, biomass of new forest plantations is generally lower than the biomass of an old one (e.g. a forest that is typically harvested).

The obtained maps of $\rho_{loss}$ and $\rho_{gain}$ in Eqs. (1) and (2) refer to sparse and isolated pixels where there have been forest gain or loss. To obtain continuous fields, such density values were spatialized by computing their median over a 0.1° grid. Annual maps of biomass were finally obtained at 100 m spatial resolution as follows:

$$B_t = B_{2010} + \alpha \cdot \rho_{loss} \cdot TC_{t,loss} - \rho_{gain} \cdot TC_{t,gain} \cdot \frac{(2010-t)}{10}, \tag{4}$$

where $t$ is the year (over the 2000–2017 period) and $\alpha$ takes the value of $+1$ for $t < 2009$, and $-1$ otherwise. $TC_{t,loss}$ and $TC_{t,gain}$ are derived following the above-described approach for year 2010. The analysis was implemented in Google Earth Engine[64] to efficiently handle the large data archives.

**Biomass losses due to natural disturbances.** We expressed forest vulnerability as the relative biomass loss following the occurrence of a given natural disturbance. For each disturbed forest area at year $t$—recorded in the disturbance databases (EFFIS, FORWIND and IDS-USDA)—the corresponding relative biomass loss ($BL_{rel}$) was quantified based on the difference between pre- and post-disturbance biomass ($B$) (Supplementary Fig. 1, step 2.1), as follows:

$$BL_{rel} = \left[ \frac{\max(B_{t-n}, \dots, B_t) - \min(B_t, \dots, B_{t+m})}{\max(B_{t-n}, \dots, B_t)} \right], \tag{5}$$

where $n$ and $m$ represent the backward and forward time lags (in years), respectively, and express the time window over which a biomass loss can be reasonably attributed to a given disturbance. For fires and windthrows, $n$ and $m$ were both set to 1, as these disturbances typically lead to an abrupt loss in vegetation. For insect outbreaks, $n$ and $m$ were set to 2 and 5, respectively, in order to capture the progressive and slow change in biomass following an insect infestation[53]. Input data in Eq. (5) were obtained by spatially averaging the values of annual biomass maps over the disturbed forest patch. $BL_{rel}$ represents the response variable in our vulnerability modelling framework.

**Environmental predictors and PFTs.** The estimate of $BL_{rel}$ was complemented with a set of environmental variables of three major categories: forest (F), climate (C) and landscape (L) features selected as potential predictors of forest vulnerability based on existing literature (Supplementary Fig. 1, step 2.2). These variables were collected from multiple sources, including satellite and reanalysis products (Supplementary Methods 1) and spatially averaged over the forest area of each disturbance record. Forest features include vegetation parameters describing the forest state and productivity, such as above ground biomass, leaf area index (LAI), tree age, tree density and tree height. Climate features include annual values of temperature, precipitation and snow conditions, their long-term averages, and their anomalies in the years preceding the disturbance, as well as extreme event

indicators such as standardized precipitation evapotranspiration index SPEI-12, moisture index and maximum wind speeds. Landscape features include population density, spatial variability of landscape patterns and geomorphological parameters. Such multi-variate approach enabled us to account for possible amplification or dampening effects among multiple susceptibility drivers, which are typical of compound events[29]. Environmental variables have annual temporal resolution (dynamic layers) or represent climatological values of the entire observational period or of a specific era/year (static layers) (see Supplementary Methods 1). Other variables, such as tree species composition and diversity, not included explicitly in our assessment may affect vulnerability as well[65]. However, the lack of a systematic monitoring of these quantities hampered their integration in our large-scale assessment.

Finally, for each observed damaged area, the cover fractions of different PFTs were retrieved from the landcover maps of the European Space Agency Climate Change Initiative[66] (ESA-CCI, https://www.esa-landcover-cci.org/) including broadleaved deciduous (BrDe), broadleaved evergreen (BrEv), needle leaf deciduous (NeDe) and needle leaf evergreen (NeEv).

**Vulnerability modelling.** Quantifying the risks for forest ecosystem services due to natural disturbances requires the integration of hazard, exposure and vulnerability components[32,33]. Hazard represents the occurrence of the agent affecting the forest ecosystems (e.g. insect pest outbreak); exposure refers to the distribution of forest ecosystem services potentially prone to a hazard; and vulnerability expresses the degree to which a forest ecosystem is affected when exposed to a given disturbance. In this study we focus on the vulnerability component quantified in terms of relative biomass losses ($BL_{rel}$) following the occurrence of a specific hazard (0% means a forest is not vulnerable to the given disturbance, 100% means a forest is completely damaged when exposed to the given disturbance). Therefore, our estimates should not be confused with the overall risk levels, which incorporate in addition to vulnerability also the probability of occurrence of disturbance and the exposure[32,33].

For each disturbance type, we developed an RF regression model[31] to predict the observed $BL_{rel}$ (response variable) based on pre-event environmental conditions (predictors). The use of machine learning in general and of RF in particular, being nonparametric and nonlinear data-driven methods, avoids making potentially strong assumptions about the functional form relating the key drivers and the response functions to natural disturbances.

First, in order to focus on effective damaging events in forest ecosystems, only records with $BL_{rel} > 5\%$ were selected (Supplementary Fig. 1, step 3). In the case of windthrows, we noted that maximum wind speeds retrieved from 0.5° spatial resolution of reanalysis data may largely underestimate effective maximum winds. This was particularly evident for tornado events, given their limited spatial extents compared to the grid cell, and the storm event Klaus that occurred in 2009 and for which we noticed an underestimation of the effective wind speed of the 78% (retrieved ~12 ms$^{-1}$ instead of observed maximum wind speed of 55 ms$^{-1}$ (ref. [67]). Therefore, such events were excluded from our analysis.

Possible missing data in the environmental variables were corrected by the median value of the variable-specific distributions (Supplementary Fig. 1, step 4). Potential effects of spatial dependence structure in the observational datasets were reduced by resampling $BL_{rel}$, F, C and L along the gradients of the three principal components (PC) derived from the initial set of predictors. To this aim, we used 20 bins of equal intervals for each PC dimension spanning the full range of values. The resampling procedure was stratified by splitting the records in training and testing sets. For each year between 2000 and 2017, we randomly extracted 60% of the records. The extracted subset ($BL_{rel}$, F, C and L) was then binned in the PC space using the average as aggregation metric weighted by the areal extents of each disturbance record. The remaining 40% of records were similarly processed and used as a separate validation set (Supplementary Fig. 1, steps 5–7). The cover fraction of each PFT was resampled using the same approach and renormalized within each bin. Only bins with at least three records were retained for model development.

The resampled training and testing sets were used to calibrate and validate an "approximate" RF model using the full set of variables (A) as predictors initially identified based on literature review (Supplementary Fig. 1, step 8 and Supplementary Table 1). With the RF algorithm importance scores for each environmental variable can be calculated[31]. These scores reflect how important each covariate is in determining the fitted values of relative biomass loss. The RF implemented here uses 500 regression trees, whose depth and number of predictors to sample at each node were identified using Bayesian optimization. To reduce potential redundancy effects across predictors and facilitate the interpretability of results, we implemented a feature selection procedure. Based on the "approximate" RF model the importance of each predictor was quantified. We then computed the Spearman correlation between each pair of predictors and when it exceeded 0.8, the predictor with the lower variable importance was excluded (Supplementary Fig. 1, step 9 and Supplementary Table 1). The remaining predictors (I) were then used for a second set of RF runs, in which we iteratively evaluated RF performance on a reduced set of predictors, excluding in each new run the less important variable computed on the new reduced set of features. The set of predictors which maximizes the $R^2$ was finally selected (Q hereafter for short) (Supplementary Fig. 1, step 10 and Supplementary Table 1). The implemented iterative feature selection

procedure identifies a reasonable compromise between computing cost and model performance. The general equation describing the vulnerability is as follows:

$$BL_{rel} = \nu(Q),  \quad (6)$$

where $\nu$ is the vulnerability model implemented in the RF regression algorithm, and describes the relative biomass losses as a function of a selected $Q$ set of environmental variables.

Such automatic feature selection process was complemented with visual interpretation of the PDPs[68] based on the RF algorithm. PDP is used to visualize the relationship between explanatory covariates (environmental predictors) and $BL_{rel}$, independent of other covariates (Supplementary Figs. 2–4). PDP results were analysed in combination with a detailed study of the literature and allowed us to understand and interpret the response functions to natural disturbances (see details in the main text and Fig. 2). Consistency of PDPs at the boundaries of the observational ranges was carefully checked to reduce possible artefacts generated when the models are used to extrapolate outside the range of training conditions.

Vulnerability models were further refined by retrieving $\nu$ functions separately for each PFT. For PFT-specific vulnerability models, only resampled records in the PC space with a cover fraction >5% were retained and used for the model development (Supplementary Fig. 1, step 11). Model performances were ultimately evaluated on the testing set in terms of coefficient of determination ($R^2$), root mean square error (RMSE), percent bias (PBIAS)[69] and RE.

Regarding the insect-related disturbance, we initially implemented specific RF models for different insect groups (bark beetles, sucking insect and defoliators). However, due to the limited sample size of the first two groups, RF was not able to represent their effects on biomass losses reliably. We therefore opted to merge all three groups in a unique insect disturbance class (hereafter referred as insect outbreaks). We recognize that different ecological processes may characterize each insect group and therefore the use of a unique insect class may potentially mask some distinctive features. The resulting vulnerability models can therefore identify only drivers and patterns common to all groups (e.g., susceptibility to temperature anomalies[70,71]).

**Interacting processes**. The co-occurrence of multi-dimensional environmental factors resulting from the combination of interacting physical processes (compound events) may amplify or dampen ecosystem responses[29]. Tree-based models consider all variables together in the model and account for nonlinear feature interactions in the final model[31,68]. The inherent ability of RF models to detect interacting variables allows avoiding the prescription of specific relations between variables based on "a priori" knowledge—as for instance required in parametric regression frameworks—by letting the model learn automatically these relations from data.

In order to detect feature interactions and assess their strength in the developed RF-based vulnerability models we computed the Friedman's $H$-statistic[50]. Here, we derived the $H$-statistic to assess second-order interactions by quantifying how much of the variation of the prediction depends on two-way interactions. To speed up the computation, we sampled 50 equally spaced data points over the environmental gradients.

We complemented this analysis by estimating the amplification or dampening effect ($\Delta P$) associated to each feature interaction. To this aim, we quantified the difference in the peak values between the response function which incorporates interacting processes (two-way partial dependences) and those ones decomposed without interactions (one-dimensional partial dependences) and expressed in terms of relative variations.

The $H$ and $\Delta P$ metrics were computed for each pair of features, and averaged for different combinations of predictor categories (forest, climate, landscape).

**Spatial and temporal patterns of vulnerability and its key drivers**. The RF models were used to evaluate the vulnerability of forests annually between 1979 and 2018 for each grid cell (0.25°) of the spatial domain covering the geographic Europe (including Turkey and European Russia). To this aim, vulnerability models were used in predictive mode using as input spatial maps of predictors, preliminary resampled to the common resolution, and with results expressed in terms of potential relative biomass loss ($PBL_{rel}$). Estimates of $PBL_{rel}$ are obtained as the average from all trees in the RF ensemble. The ongoing changes in climate features were also accounted for in our framework. Climate predictors were kept dynamic for backward RF runs, while the remaining forest and landscape features were fixed to their current values averaged over the 2009–2018 period. Doing so, we implicitly assume that the sampling of response variables and predictors is representative for the whole temporal period. However, over longer time periods (from decades to century) additional ecosystem processes may play a role, such as adaptation phenomena driven by species change and shifting biomes, which could also affect vulnerability trends. The lack of multi-temporal monitoring of most of the forest and landscape predictors hampered the integration of their dynamics in the backward RF runs.

Results of PFT-specific vulnerability models were averaged at grid-cell level with weighting based on the cover fractions of PFTs (Supplementary Fig. 1, steps 12–13). This resulted in annual maps of vulnerability to each natural disturbance. Spatial and temporal variations in vulnerability were both expressed in relative and absolute terms. Absolute biomass losses were retrieved by multiplying estimates of

potential relative biomass loss by the available biomass. Therefore, vulnerability values in a given grid cell reflect the biomass (relative or absolute) that would be affected if exposed to a disturbance under its specific local and temporal environmental conditions.

Grid-cell uncertainty of predicted vulnerability values were quantified in terms of standard error (SE) derived by dividing standard deviations of the computed responses over the ensemble of the grown trees of the model by the square root of the ensemble size (Supplementary Fig. 7).

We then calculated the "current" vulnerability as the average vulnerability over the 2009–2018 period. To factor out the local dependence of the current vulnerability on each predictor we retrieved the Individual Conditional Expectation[72] (ICE) for each grid cell. ICE plots show the relationship between the predicted target variable ($PBL_{rel}$) and one predictor variable for individual cases of the predictor dataset. In our application, an individual case is a specific combination of F, C and L data for a given grid cell. To summarize and map the ICE of each grid cell in a single number, we fitted by linear regression the partial dependence of $PBL_{rel}$ versus the corresponding predictor variable and mapped the slope of this regression, hereafter referred as "local sensitivity" (Supplementary Figs. 5–7), similarly to the approach presented in ref. [30]. The marginal contribution ($Z_{marg}$) of each environmental category of predictors (F, C and L, hereafter referred as X for short) on the current vulnerability was derived as follows:

$$Z_{marg,X} = 100 \times \frac{\sum_{i \in X} |s_i|}{\sum_{j \in Q} |s_j|},  \quad (7)$$

where $s$ represents the slope of ICE, $i$ runs over all predictors of $X$, whereas $j$ runs over all available predictors $Q$. Therefore $Z_{marg,X}$ values range between 0 (no dependence of current vulnerability on $X$ predictors) and 100% (full dependence of current vulnerability on $X$ predictors).

Long-term linear trends in vulnerability ($\delta PBL_{rel}$) were quantified over the 1979–2018 period for each grid cell and their significance evaluated by the two-sided Mann–Kendall test. In order to isolate the key determinants of the emerging trends in vulnerability, a set of factorial simulations was performed. To this aim, we estimated the vulnerability due to the temporal variations in a given $k$ climate predictor ($PBL_{rel}^k$), by applying the RF models to a data array in which the $k$ climate variable is dynamic while all the remaining features are kept fixed to their "current" value (average value over 2009–2018). The resulting trends in vulnerability associated to the $k$ factor ($PBL_{rel}^k$) are then calculated by linear regression and subject to the Mann–Kendall test.

Spatial and temporal patterns were visualized at grid-point scale and averaged over geographic macro-regions (Supplementary Fig. 14 and Supplementary Tables 2 and 3). Zonal statistics were obtained by averaging grid-cell results weighted by their forest areal extent. Forests with cover fraction lower than 0.1 were excluded from the analyses. Uncertainty in spatial averages were based on the 95% bootstrap confidence interval computed with 100 bootstrap samples.

In order to derive statistics minimally affected by potential extrapolation errors of the RF models, we replicated the aforementioned analyses by excluding areas outside the observational ranges of climatological temperature and precipitation (Supplementary Fig. 8).

**Combining forest vulnerability to multiple natural disturbances**. To quantify the total vulnerability to multiple disturbances we defined the OVI, similarly to the multi-hazard index developed in ref. [73]. We assumed that the considered disturbances are independent and mutually non-exclusive and the potential biomass loss of single disturbances is spread homogeneously within each grid cell. From the inclusion-exclusion principle of combinatorics the potential biomass loss associated to the OVI can be expressed for a given year as follows:

$$PBL_{rel}(OVI) = \bigcup_{p=1}^{D} PBL_{rel,p} = \sum_{q=1}^{D} \left( (-1)^{q-1} \cdot \sum_{\substack{G \subset \{1,\ldots,D\} \\ |G|=q}} PBL_{rel,G} \right),  \quad (8)$$

where $p$ refers to the disturbance-specific $PBL_{rel}$, $D$ is the number of disturbances considered, the last sum runs over all subsets $G$ of the indices $\{1, \ldots, D\}$ containing exactly $q$ elements, and

$$PBL_{rel,G} := \bigcap_{p \in I} PBL_{rel,p},  \quad (9)$$

expresses the intersection of all those $PBL_{rel,p}$ with index in $G$. Maps of current overall vulnerability and trends were ultimately analysed following the approach adopted for single disturbances.

This approach does not account for the potential reduction in exposed biomass following the occurrence of a given disturbance. Furthermore, possible amplification/dampening effects due to interacting disturbances could also occur[3,74]. A strong interaction effect has been documented for instance between windthrows and bark beetle disturbances. Uprooted trees are virtually defenseless breeding material supporting the build-up of beetle populations and the consequent increase in vulnerability to insect outbreaks[3,59]. Insect outbreaks, in turn, may potentially affect the severity of subsequent forest fires by altering the abundance of available fuel[60]. The magnitude of these effects varies with insect type

and outbreak timing. Despite the relevance of these interactions, the lack of reference observational data of compound events hampered the integration of their effects in our modelling framework. Therefore, estimates of OVI can only partially capture the overall vulnerability resulting from multiple disturbances and should be viewed in light of these limitations.

Spatial maps of current overall vulnerability and trends in OVI were then normalized separately based on the min–max method and combined by simple multiplication into a single index, hereafter referred as space-time integrated OVI. High values of space-time integrated OVI depict forest areas that are currently susceptible to multiple disturbances and their vulnerability have experienced a substantial increase over the 1979–2018 period. The space-time integrated OVI is used to identify currently fragile ecosystems that might in the future become even more susceptible to natural disturbances.

**Reporting summary**. Further information on research design is available in the Nature Research Reporting Summary linked to this article.

## Data availability

The observation-driven datasets analysed in this study are publicly available as referenced within the article (Methods and Supplementary Methods 1). The generated vulnerability models are available at https://doi.org/10.6084/m9.figshare.13577960.

## Code availability

The custom MATLAB code written to analyse the data, develop the random forest models and generate figures is available at https://doi.org/10.6084/m9.figshare.13577960. Additional codes written in R/Python and Google Earth Engine used for data pre-processing are available on request from the corresponding author.

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

## Acknowledgements

The study was funded by the Exploratory Project FOREST@RISK of the European Commission, Joint Research Centre. G. Camps-Valls was supported by the European Research Council (ERC) through the ERC Consolidator Grant SEDAL (project id 647423). The views expressed are purely those of the writers and may in no circumstance be regarded as stating an official position of the European Commission.

## Author contributions

G.F. designed the study and developed the vulnerability models; M.G. and J.S. assisted in data integration tasks; G. Ceccherini computed annual biomass losses. All authors contributed to interpreting the results and writing the manuscript.

## Competing interests

The authors declare no competing interests.
