## [Peer Review File · Nature Communications]

Reviewer comments, first round:

Reviewer #2 (Remarks to the Author):

I have found this to be a very well written and organized manuscript. The topic of forest vulnerability in light of climate change is timely, and the authors have done very well to extract insightful findings from geospatial data. The use of Random Forests to extract trends from the large data set is very helpful and the predictive maps of vulnerability are well-presented.

Despite the positives, noted above, there are several concerns that I have that require attention:

1 - I am confused about how the USDA Insect and Disease survey data were incorporated into the analysis. I understand that you used a European wildfire and windthrow databases, but it is not clear how you incorporated the database from the US to calibrate an empirical random forest model in Europe. This lack of clarity undermines the important message that the authors wish to convey

2 - I am confused about how a change in biomass was calculated and how the parameterization of biomass change was determined. The text is not at all clear - so I cannot fully evaluate the worthiness of this manuscript for publication when I do not know how the dependent variable was created/parameterized

3 - It is unclear how the authors thought about interacting disturbances. Insect outbreaks can provoke fires, for example (and insect outbreaks can provoke windthrow). Given the importance of interactive effects, the manuscript is weakened when it does not attend to them

4 - Taking a look at the results - how can the area vulnerable to windthrow be equal to the area vulnerable to wildfire? That seems quite illogical to me. Windthrow damage events are less frequent and cover much smaller areas than wildfire.

5 - In the figures the authors use acronyms to represent variables. I could not properly follow what the variables were. This weakens the manuscript a great deal.

6 - Why are variables representing "static" factors such as forest age and density more important than dynamic variables such as temperature and precipitation? A wildfire will be ignited and burn really any type of forest, predicated by the weather conditions of the day/week. Therefore, this makes me question how the data were prepared. For example -if a fire occurred on July 1st in a given year, how was that linked to the corresponding climate information? Is the record string based on a day or several days? This is all important because it impacts how the model is being conceptualized.

7 - Forest management scenarios

What is the theory/conceptualization to justify a scenario of -15% +15% tree density and age? Where is the supporting published justification for this? It seems to be a bit of a wild guess.

Additionally, related to my earlier point about the importance of dynamic variables -- it is largely proven that climate is the much more dominant factor in wildfire risk. Modifying local climate to mitigate wildfire risk is a much taller order. Also, changing the density of forest will not really help with wildfire or insect outbreak at the regional scale. There are many conceptual issues with this manuscript that lead to a weakening of the tractability of the overall results, unfortunately.

Reviewer #3 (Remarks to the Author):

The authors develop RF models to characterize the drivers of fire, windthrow, and insect outbreaks across European forests during 2000-2017, and then apply these RF models to assess current forest vulnerability and the contributions of changes in climate to vulnerability over 1979-2017.

They stress vulnerability is defined as the biomass loss expected given a disturbance event occurs, not the likelihood of an event occurring. Based on variables describing forest condition that were important in the RF models, they develop scenarios of forest management to assess the potential impacts on forest vulnerability. They conclude that forest properties have the largest effect on current forest vulnerability to fire and windthrow, and that rising temperatures have increased forest vulnerability to insect outbreaks. They suggest their results can be used to benchmark land surface models and inform forest management practices to reduce vulnerability.

The paper is well written, with a clear description of the methods that allows for reproducibility. They have created excellent figures that clearly convey their results (but see suggestions for improving the response plots). However, I have multiple questions about the response functions in the RF models where some responses appear to be counter-intuitive. These questions are noted with review comments in the Word document. In many instances the MAD around the mean response spans 0, indicating the data are insufficient to establish a conclusive effect. Also, there is not an evaluation of the RF models. I am concerned with how these issues influence the interpretation of the model goodness of fit. I am therefore unable to assess the model application for predictions of forest vulnerability across Europe, the potential impacts of management scenarios, and the relative contribution of climate compared with forest and landscape characteristics.

In addition to my specific comments in the Word document, I have these over-arching suggestions:

1. Include interaction terms in the RF models. There are several places where interactions between predictors are implied but not evaluated. E.g. " Higher average temperature (avg aTavg) appears a key driver of forest vulnerability to insect outbreaks, particularly when combined with droughts (avg aPcum and avg SPEI), possibly because heat and water stress reduce plant resistance to pest damage". Including interaction terms may also affect the interpretation of the influence of those predictors on forest vulnerability.
2. Include a thorough evaluation, e.g. table of classification accuracy, plots of observed vs. predicted biomass loss, plots of predictors vs. residuals, for each RF model. This will allow the reader to come to their own conclusions about goodness of fit.
3. Include a discussion of how model uncertainty and lack of fit to observations affects the predicted forest vulnerability and responses to management. For example, it's curious that in the model of fire vulnerability, biomass loss declines at the lowest MI. I would expect more biomass loss to fire in the driest areas. Where in Europe does the lowest MI occur? Is there a confounding influence? Is pre-fire biomass very low in these areas, resulting in nothing to burn? How might this decline at low MI be influencing the results of the model application? Use where the model is unexpected, uncertain, or wrong to gain more insight into the processes that may be driving biomass loss.

I appreciate the large amount of thought and work that went into this paper. The scale of the analysis is impressive and the methods are well suited to the goals. Including these revisions might allow for more robust conclusions as to current forest vulnerability and implications for forest management. I agree that such information would be helpful for land surface model benchmarking and guiding management practices, both of which are much needed.

First of all, we would like to thank the two referees for their insightful and constructive comments. In our revision of the manuscript we tried to address all their comments and suggestions in order improve the robustness of the analysis and the clarity of the interpretation.

In the following, we respond to each reviewer's comment by referring to line numbers of the revised tracked version, when not differently indicated. References cited along these responses are reported at the bottom of the document.

Reviewer #2

I have found this to be a very well written and organized manuscript. The topic of forest vulnerability in light of climate change is timely, and the authors have done very well to extract insightful findings from geospatial data. The use of Random Forests to extract trends from the large data set is very helpful and the predictive maps of vulnerability are well-presented.

Despite the positives, noted above, there are several concerns that I have that require attention:

Brief premise

We thank the reviewer for her/his positive comments. Before responding to each specific concern in the following lines, we believe that a brief premise here may help clarify recurring issues raised by the reviewer in several comments. In particular, we think that the first version of our manuscript may not have provided enough detail on the vulnerability concept. This is probably the cause of confusion and misunderstanding of some of the proposed methods, which we clarified in the new version of the manuscript. The vulnerability concept has been originally presented in the IPCC report¹ and differentiates between vulnerability, exposure, hazard and risk.

According to the IPCC risk assessment framework¹, quantifying the risks for forest ecosystem services due to natural disturbances requires the integration of hazard (H), exposure (E) and vulnerability (V) components (Figure R1). Hazard represents the agent affecting the forest ecosystems; exposure refers to the distribution of forest ecosystem services potentially prone to a hazard, and vulnerability expresses the degree to which an exposed forest system is affected by a hazard. The vulnerability is therefore a metric of sensitivity of the system once the hazard takes place. The risk (R) reflects the combination of the above-mentioned components and is often expressed by multiplicative terms: $R = H \cdot E \cdot V$.

Changes in both the climate system and socio-economic processes may affect in different ways hazards, exposure and vulnerability and it is therefore important to assess these three components of climate risk independently.

This work focuses on the vulnerability (V) component only. We now better stress along the manuscript that our estimates should not be interpreted as risk levels (R) according to the IPCC framework¹. They only reflect vulnerability, i.e., the relative biomass loss conditional to a disturbance occurring, and do not integrate information on the occurrence probability of hazards (i.e. probability of observing e.g. a forest fire or a windstorms) nor on the exposure of the system to the hazard (i.e. the amount of biomass

threatened by the hazard). This is highlighted in several parts of the manuscript (e.g., in lines 83-86, lines 434-444, lines 561-563).

Figure R1. Risk of climate-related impacts results from the integration of climate-related hazards with the vulnerability and exposure of natural systems. Changes in both the climate system (left) and socio-economic processes (right) are drivers of hazards, exposure and vulnerability. Adapted from ref. (1).

1. *I am confused about how the USDA Insect and Disease survey data were incorporated into the analysis. I understand that you used a European wildfire and windthrow databases, but it is not clear how you incorporated the database from the US to calibrate an empirical random forest model in Europe. This lack of clarity undermines the important message that the authors wish to convey.*

➔ Even if the study focuses on Europe, for insect diseases we had to use the IDS-USDA database due to the lack of an analogous monitoring system for Europe. Therefore, the models of vulnerability to insect outbreaks were identified, calibrated, and validated on US data and then applied in predictive mode to Europe. To ensure transferability, we developed models for functional groups instead of working on species-specific models. For this purpose, we classified records based on functional groups of the pest (bark beetles, defoliators and sucking insects) and on the plant functional type (PFT) of the host tree species. Records were considered if the host plant belonged to the following plant functional types: broadleaved deciduous, broadleaved evergreen, needle leaf deciduous and needle leaf evergreen. The emerging patterns of forest vulnerability to insect outbreaks in Europe (see details in main text) appear robust and consistent with previous findings at local scales²⁻⁴. While such comparisons cannot be considered an empirical proof of the transferability of models, they corroborate the model suitability in the

European environmental context. We now motivate and discuss the use of IDS-USDA data in the revised version (see lines 333-340).

- To improve the situation regarding insect and disease data in Europe, we highlight that we have recently launched a joint research initiative to develop the first Database of European Forest Insect & Disease Disturbances (DEFID2) (<https://forest.jrc.ec.europa.eu/en/>). Once available, this new data will make it possible to refine the models of vulnerability to insect outbreaks for EU conditions.
2. *I am confused about how a change in biomass was calculated and how the parameterization of biomass change was determined. The text is not at all clear - so I cannot fully evaluate the worthiness of this manuscript for publication when I do not know how the dependent variable was created/parameterized.*
- We agree with the reviewer. The description of the methods missed important details to fully understand the response variable. In our initial submission, in order to fit the journal standards on the length of the manuscript, we have severely shortened the text. In this revised version, we have included a new section in Methods to describe the approach used to retrieve multi-temporal biomass data (see section “Reconstruction of annual biomass time series”) and added a new supplementary figure (Supplementary Fig. 13).
3. *It is unclear how the authors thought about interacting disturbances. Insect outbreaks can provoke fires, for example (and insect outbreaks can provoke windthrow). Given the importance of interactive effects, the manuscript is weakened when it does not attend to them*
- We point out that our study is limited to the assessment of vulnerability and therefore does not address the triggering processes of natural disturbances. In fact, the important interacting mechanisms mentioned by the reviewer characterize the probability of occurrence of disturbance events and, therefore, refer to the hazard component of the risk framework proposed by the IPCC¹ and adopted here (please, see also “brief premise” above). The characterization of triggering processes is beyond the scope of this study. However, we recognize that interacting processes might also affect the degree of vulnerability of forest to natural disturbances, thus the vulnerability component that is the focus of this work.
 - We agree with the reviewer on the potential relevance of multiple interacting disturbances. Disturbance agents are rarely independent but tend to exacerbate each other in space and time. A strong interaction effect has been documented for instance between windthrows and bark beetle disturbances. Uprooted trees are virtually defenseless against bark beetles and provide readily available breeding material that promotes the build-up of beetle populations and the consequent increase in vulnerability to insect outbreaks^{5,6}. Insect outbreaks, in turn, may potentially affect the severity of subsequent forest fires by altering the abundance of potential available fuel⁷. The magnitude of these effects varies with insect type and timing. Therefore, as changes in the climate system are conducive to an increase in disturbance regimes, an intensification of compound effects originating from multiple disturbance interactions can be expected as well⁸⁻¹⁰. Cascading and amplification effects originating from natural disturbance interactions have the potential to exacerbate the vulnerability of forest ecosystems and lead to irreversible shifts in

ecosystem states¹¹. However, to account properly for such interacting processes, large datasets of concurrent (or linked) disturbances are required, which, unfortunately, are currently missing.

- ➔ Recent research initiatives launched by the European Commission aimed at mapping natural disturbances in European Forests (e.g., Database of European Forest Insect & Disease Disturbances (DEFID2), <https://forest.jrc.ec.europa.eu/en/>) will pave the road for a quantitative assessment of such interacting processes. Given the current lack of such systematic monitoring systems of forest disturbances for Europe, in this study the unique “known” interactions explicitly considered are the effects of droughts on wildfires and insect outbreaks. However, droughts are treated in this study as a climate stressor and not a specific disturbance causing biomass loss. Therefore, our results only partially capture the effects of interacting disturbances and should be viewed in light of these limitations. We have clarified this in the text (lines 289-294 and lines 615-625).
- ➔ In addition to interactions amongst multiple disturbances, the co-occurrence of multi-dimensional environmental factors resulting from the combination of interacting physical processes may increase as well the complexity of the ecosystems response¹⁰. We have explored in detail such interacting processes resulting from the interplay between multiple environmental predictors (e.g., amplification of vulnerability due to the combination of high FWI and high biomass). We have included a new section in Methods (“Interacting processes”) and added a new figure (Fig. 3) and related discussion in the main text (lines 150-166) and in Supplementary material (Supplementary Fig. 6).

4. *Taking a look at the results - how can the area vulnerable to windthrow be equal to the area vulnerable to wildfire? That seems quite illogical to me. Windthrow damage events are less frequent and cover much smaller areas than wildfire.*

- ➔ We point out that our estimates refer to the vulnerability expressed as relative biomass loss expected IF a disturbance event occurs. They do not integrate information on area extents of the disturbance (i.e. exposure) nor the frequency of occurrence (see the “Brief premise” section on top). Our results show that 34.6 billion tonnes of biomass could be potentially destroyed due to natural disturbances, with comparable forest fraction losses associated to windthrows (38%) and fires (36%). This does not imply any conclusion on the effective risk levels (i.e. actual forest biomass loss) from windthrows and fires, which requires the vulnerability to be integrated with the hazard and exposure.
- ➔ From a general perspective, we agree with the reviewer that windthrow damage events may occur less frequently – at least for some regions of Europe – and have typically smaller area extents compared to fires. That being said, risks resulting from windthrow events recorded at country levels in the last decades have been larger than fires (see pag. 25 in ¹² and ^{8,13}). Indeed, windstorm damage events may have devastating effects especially in high biomass forests¹⁴. Therefore, our results on vulnerability are fully consistent with the reported estimates of damages.

5. *In the figures the authors use acronyms to represent variables. I could not properly follow what the variables were. This weakens the manuscript a great deal.*

- We appreciate the reviewer's comment. At the same time, we fear that describing all acronyms used in the figure captions may be redundant. We did our best to clarify further this issue by spelling the acronyms in figure captions when appropriate, and by referring to Table 1, which reports the full list of variables used in the text and figures.
6. *Why are variables representing "static" factors such as forest age and density more important than dynamic variables such as temperature and precipitation? A wildfire will be ignited on a burn really any type of forest, predicated by the weather conditions of the day/week. Therefore, this makes me question how the data were prepared. For example -if a fire occurred on July 1st in a given year, how was that linked to the corresponding climate information? Is the record string based on a day or several days? This is all important because it impacts how the model is being conceptualized.*
- We point out that in our study, we do not explore the triggering mechanisms of forest disturbances but the vulnerability of forest expressed as the biomass loss expected given a disturbance event occurs (see "brief premise" above). By referring to the example mentioned by the reviewer, our results show that – once a fire has occurred – the amount of vulnerable biomass is more strongly controlled by tree density than by average annual precipitation and temperature (Fig. 2a). In our opinion, this is an intuitive result as both fire spread and severity affect vulnerability and are strictly linked to forest structure characteristics that determine fuel availability¹⁵ (e.g., tree density). These results are further corroborated by recent studies, which found that fuel availability controls wildfire severity more than weather conditions in boreal forests¹⁶. Some of the variables representing forest structure in our study are dynamic (e.g., LAI, biomass) while others (e.g., tree density, tree age, tree height) are static because no multi-temporal data exist at European level. Thus, we implicitly have to assume in our study that such static variables are representative of the average forest structure over the observational period.
- Dynamic climate components also play an important role in affecting forest vulnerability to fires. The second and third most important variables characterizing the vulnerability to fires are the Fire Weather Index (FWI) and the Moisture Index (MI), both dynamic climatic components driven by temperature and precipitation amongst other factors (Fig. 2a). Overall, we estimated that about 37% of the vulnerability to fires in Europe (thus a substantial proportion) is controlled by dynamic climate features (Fig. 3b and lines 213-216). This confirms the key role climate plays in forest vulnerability to fires and thus corroborating the reviewer's expectations.
- Regarding the temporal aggregation, environmental variables have annual temporal resolution (dynamic layers) or represent climatological values of the entire observational period or of a specific era/year (static layers). More details are provided in Supplementary Text S1. Changes in environmental conditions occurring over shorter temporal scales (e.g., seasonal/weekly/daily) may also affect forest vulnerability; however, to properly capture these effects, shorter-term biomass variations – currently missing at the appropriate spatial resolution – would be needed. Therefore, the developed vulnerability models are suited input components in a risk framework designed for annual or longer temporal scale analyses. We have further clarified this in the text (lines 422-424) and in Table 1.

7. *Forest management scenarios*

(A) What is the theory/conceptualization to justify a scenario of -15% +15% tree density and age? Where is the supporting published justification for this? It seems to be a bit of a wild guess.

(B) Additionally, related to my earlier point about the importance of dynamic variables -- it is largely proven that climate is the much more dominant factor in wildfire risk. Modifying local climate to mitigate wildfire risk is a much taller order. Also, changing the density of forest will not really help with wildfire or insect outbreak at the regional scale. There are many conceptual issues with this manuscript that lead to a weakening of the tractability of the overall results, unfortunately.

- ➔ We carefully underscore that our estimates should not be interpreted as risk levels according to the IPCC framework¹ (see the “brief premise” above). Therefore, the concern (B) would be pertinent if our analysis was focused on the risks of natural disturbances. We agree with the reviewer that the climate component is likely the dominant factor of risks of natural disturbances whose triggering mechanisms are expected to be largely driven by climate (such as fires and insect outbreaks). Again, this is beyond the scope of this paper, as this study focuses solely on the vulnerability component.
- ➔ That being said, we appreciate your concern (A) regarding the range of variability of the designed forest management scenarios. They would need to be corroborated by literature and eventually adjusted to “real” forest management scenarios. The section on forest management may have reduced both clarity and focus of the paper and we have decided to remove this section. We will explore the effect of forest management scenarios more comprehensively in a future contribution, when the analysis of hazard and exposure will be integrated in a comprehensive risk assessment framework.

Side remark: please note that uncertainty bounds in figure 5 have been corrected.

Reviewer #3

The authors develop RF models to characterize the drivers of fire, windthrow, and insect outbreaks across European forests during 2000-2017, and then apply these RF models to assess current forest vulnerability and the contributions of changes in climate to vulnerability over 1979-2017. They stress vulnerability is defined as the biomass loss expected given a disturbance event occurs, not the likelihood of an event occurring. Based on variables describing forest condition that were important in the RF models, they develop scenarios of forest management to assess the potential impacts on forest vulnerability. They conclude that forest properties have the largest effect on current forest vulnerability to fire and windthrow, and that rising temperatures have increased forest vulnerability to insect outbreaks. They suggest their results can be used to benchmark land surface models and inform forest management practices to reduce vulnerability.

The paper is well written, with a clear description of the methods that allows for reproducibility. They have created excellent figures that clearly convey their results (but see suggestions for improving the response plots). However, I have multiple questions about the response functions in the RF models where some responses appear to be counter-intuitive. These questions are noted with review comments in the Word document.

- 1. In many instances the MAD around the mean response spans 0, indicating the data are insufficient to establish a conclusive effect.*
- 2. Also, there is not an evaluation of the RF models. I am concerned with how these issues influence the interpretation of the model goodness of fit. I am therefore unable to assess the model application for predictions of forest vulnerability across Europe, the potential impacts of management scenarios, and the relative contribution of climate compared with forest and landscape characteristics.*

We thank the reviewer for her/his positive comments. We first address here briefly the two major concerns raised by the reviewer, and then go into more detail in the specific responses. We believe that the first version of the manuscript may have suffered a lack of detail which ultimately led to confusion and misunderstanding of our results/figures:

- ➔ (1) Regarding the MAD around the mean response spanning 0. The reviewer refers to Fig. 1d-f and Supplementary Figs. 2-4 (now Fig. 2d-f and Supplementary Figs. 3-5 in this revised version). For convenience response functions are visualized by zero-centered partial dependence plots. This does not mean that the actual response function is close to zero or negligible. The centered partial dependence is offset so that all individual conditional expectations start from zero to facilitate the examination of the cumulative effect of the selected feature¹⁷. To clarify this, we have added the corresponding offset values in each panel.
- ➔ (2) Regarding the evaluation of RF models. Model performances were properly evaluated using standard methodology applied in the field of machine learning. In particular, we have implemented a conventional approach for calibration and validation processes (60% of the records used for calibration and the remaining 40% for validation, lines 464-469). Model performances were assessed in terms of R^2 and shown in inset box of Fig. 1 and discussed in the main text (old manuscript version). We have further clarified this and expanded the validation to

additional performance metrics (RMSE, PBIAS, RE, lines 508-510). We have included two additional figures (Fig. 1 and Supplementary Fig. 2) and material (Section “Model evaluation”, lines 88-104). Please, see dedicated responses below.

Major comments

3. *Include interaction terms in the RF models. There are several places where interactions between predictors are implied but not evaluated. E.g. " Higher average temperature (avg aTavg) appears a key driver of forest vulnerability to insect outbreaks, particularly when combined with droughts (avg aPcum and avg SPEI), possibly because heat and water stress reduce plant resistance to pest damage". Including interaction terms may also affect the interpretation of the influence of those predictors on forest vulnerability.*

We agree with the reviewer on the relevance of interacting processes for vulnerability assessment. The co-occurrence of multi-dimensional environmental factors resulting from the combination of interacting physical processes (compound events) may indeed amplify or dampen ecosystem responses¹⁰.

Data-driven and ML models exploit multivariate data and consider the interactions between the considered input covariates, without imposing a specific parametric in the model^{18,19}. Depending on the ML model, different levels and types of interactions are accounted for. In particular, RF build and optimally combine a number of decision trees constructed using random subsets of covariates; by doing so, the model explores different (many) regions of the space and correlations between covariates which are ultimately combined optimally in terms of information to do predictions. On the downside, the relationships learned by a ML model can be complicated to interpret, but as we have shown here, there are techniques that can uncover them efficiently.

In order to detect feature interactions and assess their strength in the developed RF-based vulnerability models, we performed a specific analysis based on the H-statistic²⁰ in the revised version of the manuscript. Here, we derived H-statistic to assess second-order interactions by quantifying how much of the variation of the prediction depends on two-way interactions. If two features do not interact, their 2-way partial dependence function can be ideally decomposed as follows (assuming the partial dependence functions are centered at zero):

$$PD_{jk}(Q_j, Q_k) = PD_j(Q_j) + PD_k(Q_k)$$

where $PD_{jk}(Q_j, Q_k)$ is the 2-way partial dependence function of both features and $PD_j(Q_j)$ and $PD_k(Q_k)$ the partial dependence functions of the single features. This decomposition expresses the partial dependence function without interactions (between features j and k). The H-statistic reflects the difference between the observed 2-way partial dependence function and the decomposed one without interactions and expressed in terms of variance as follows:

$$H_{jk} = 100 \cdot \sqrt{\sum_{i=1}^N \left[PD_{jk}(Q_j^{(i)}, Q_k^{(i)}) - PD_j(Q_j^{(i)}) - PD_k(Q_k^{(i)}) \right]^2 / PD_{jk}^2(Q_j^{(i)}, Q_k^{(i)})}$$

where N represent the data points. To speed up the computation, we sampled 50 data points equally spaced over the environmental gradients. The H-statistic is 0 if there is no interaction at all and 100% if all of the variance of $PD_{jk}(Q_j, Q_k)$ is explained by the interaction term.

We complemented this analysis by quantifying the amplification or dampening effect associated with each feature interaction. To this aim, we quantified the difference in the peak values between the observed 2-way partial dependence function and the decomposed one without interactions and expressed in terms of variance as follows:

$$\Delta P_{jk} = 100 \cdot \left[\max \left(PD_{jk}(Q_j, Q_k) \right) - \max \left(PD_j(Q_j), PD_k(Q_k) \right) \right] / \max \left(PD_j(Q_j), PD_k(Q_k) \right)$$

Positive values indicate an amplification effect originating from the interaction of the two features, while negative values reflect a dampening effect. The magnitude of the amplification/dampening effect is expressed in relative terms with respect to the maximum value observed in the single-feature partial dependence functions.

The H_{jk} and ΔP_{jk} metrics are computed for each pair of j - k features, and aggregated for different combinations of predictor categories (forest, climate, landscape).

- ➔ We have included a new section in Methods (“Interacting processes”) to describe the above-mentioned methodology and results are discussed in the text (lines 150-166). Two new figures have been added to the revised version (Fig. 3 and Supplementary Fig. 6).

- 4. *Include a thorough evaluation, e.g. table of classification accuracy, plots of observed vs. predicted biomass loss, plots of predictors vs. residuals, for each RF model. This will allow the reader to come to their own conclusions about goodness of fit.*

- ➔ We have implemented a conventional approach for calibration and validation processes. We randomly extracted 60% of the records used for calibration of our models and the remaining 40% for validation, as also described in Methods (lines 464-469). Model performances were evaluated in terms of coefficient of determination (R^2), one of the most common metric used to assess the goodness of fit of a model. Results of model validation were shown in Fig. 1 (inset box) and discussed in the main text in the first section (old manuscript version).
- ➔ We have further expanded model evaluation and included additional error metrics: root mean square error (RMSE), percent bias (PBIAS) and relative error (RE) (see lines 508-510). We have dedicated a new section in the main text to better present results of model evaluation (see lines 88-104) and added two new figures (Fig. 1 and Supplementary Fig. 2).
- ➔ Regarding the reviewer’s suggestion of including a table of classification accuracy, we point out that this cannot be evaluated, as the machine-learning algorithm used for the analysis does not deal with a classification task but with a regression task.

- 5. *Include a discussion of how model uncertainty and lack of fit to observations affects the predicted forest vulnerability and responses to management. For example, it's curious that in the model of fire vulnerability, biomass loss declines at the lowest MI. I would expect more biomass loss to fire in the driest areas. Where in Europe does the lowest MI occur? Is there a confounding influence? Is pre-fire biomass very low in these areas, resulting in nothing to burn? How might this decline at low MI be influencing the results of the model application? Use where the model is unexpected, uncertain, or wrong to gain more insight into the processes that may be driving biomass loss.*

- Partial dependence plots (PDPs) are used to visualize the relationship between explanatory covariates and BL_{rel} , independent of other covariates. PDP is a powerful model-agnostic method to show whether the relationship between the target and a predictor feature is linear, monotonic or arbitrarily nonlinear. However, the marginal effect of a covariate on the response variable, so BL_{rel} is only interpretable within and not across covariates. Indeed, no interactions amongst multiple predictors are visualized in the (one-dimensional) PDPs. We have clarified this in text (see lines 150-151 and in lines 497-499).
- Regarding the specific case mentioned by reviewer on the relation between MI and vulnerability at low MI values. We confirm that the interpretation of the reviewer is correct. At low MI values there is a very low stand biomass, thus a limited amount of potential available fuel. This results in a low vulnerability at low MI values. This is evident in Fig. R2 below, where box plots of biomass values are computed over a gradient of MI. These interaction terms cannot be visualized in 1-dimensional PDPs as above-mentioned, but they are explicitly included in our RF models (see our response to your comment #3).
- We would like to point out that by definition, machine learning methods are not based on the mechanistic representation of the phenomena but encode nonlinear feature relations, and therefore it is not trivial to gain insight about the underlying processes influencing the system response to drivers. However, in recent years, the field of model interpretability has evolved to a quite mature field in machine learning, and now many techniques are available to derive feature ranking from complex ML models and to identify what features drive the prediction process inside the model. The model-agnostic methods, as those applied in this study, can provide important insights on the outputs of RF models, however their interpretation should not be assimilated as that one typically following the application of physical-based models. In physically-based models, results and response functions follow the application of a series of mathematical equations whose formulation has been theoretically assumed or tested with some sort of validation. In machine learning models, results and response functions originate directly from the observations and therefore, may potentially reveal unexpected patterns and capture local-scale effects that do not necessarily translate into a set of simple parametric equations easy to interpret but come as feature ranking, uncertainty estimates and sample relevance for the ML decision making.. We have further clarified this in text (see lines 106-109).

Figure R2. Variability of biomass values across a gradient of moisture index as recorded in the EFFIS database. The tops and bottoms of each “box” are the 25th and 75th percentiles of the samples, respectively. The line in the middle of each box is the sample median. Whiskers are drawn from the ends of the interquartile ranges to the furthest observations within the whisker length. Observations beyond the whisker length are marked as outliers here defined as a point that is more than 1.5 times the interquartile range away from the top or bottom of the box.

→ Following the reviewer’s suggestion we have computed local uncertainty in forest vulnerability predictions. Local uncertainty is expressed in terms of standard errors derived by dividing standard deviations of the computed responses over the ensemble of the trees by the square root of the ensemble size. Furthermore, in order to derive statistics minimally affected by potential extrapolation error of RF models, all spatial statistics were recomputed by excluding areas outside the observational ranges of climatological temperature and precipitation (Supplementary Fig. 8). We found that spatial statistics of vulnerability computed only in areas with climatological conditions analogous to those of the observational datasets (Methods, Supplementary Fig. 8) showed marginal variations (<1 percentage point when averaged across Europe) compared to the estimates derived from the entire spatial domain (Supplementary Table 2). These results corroborate the robustness of our findings. These new analyses have been described in Methods (lines 564-567 and lines 597-599), and shown in the new Supplementary Figs. 7 and 8 and Tables 2 and 3. An in-depth discussion on local uncertainty is now also presented in the main text (see lines 184-204).

6. *I appreciate the large amount of thought and work that went into this paper. The scale of the analysis is impressive and the methods are well suited to the goals. Including these revisions might allow for more robust conclusions as to current forest vulnerability and implications for forest management. I agree that such information would be helpful for land surface model benchmarking and guiding management practices, both of which are much needed.*

→ We thank again the reviewer for her/his positive comments. We highlight that we decided to remove the section on forest management strategies due to some concerns raised by another reviewer. Despite the fact that we consider our analysis on forest management strategies both important and correct we also recognize that an evaluation of the effectiveness of forest management strategies exclusively based on vulnerability may lead to potential misconception of the paper's main messages. We will explore our study's implications on forest management more comprehensively in a future contribution, once the analysis of hazard and exposure has been integrated in a comprehensive risk assessment framework.

Specific comments

L89: This needs more explanation for those not familiar with RF methods.

→ According to the reviewer's suggestion we have clarified this in the text.

L97-98: Your data do not show the connection to fuel availability, only to vulnerability.

→ We have rephrased the statement: "In accordance with previous studies, increased biomass, tree age and tree density, typically associated with great fuel loads, correspond to higher vulnerability to fires".

L99: what about the decline in loss at the very lowest MI?

→ Please, see our response to your major comment #5.

L102: the MAD span 0 above about 400m, so no inference is possible for higher elevations.

→ Figures show zero-centered PDPs, therefore MAD span 0 by definition. Please, see our response to your major comment #1.

L102: likely because of...

→ We have corrected the text.

L106: Except for the range between 150-200 t ha⁻¹, biomass MAD span 0, indicating no certain conclusions about its effects can be drawn.

→ Figures show zero-centered PDPs, therefore MAD span 0 by definition. Please, see our response to your major comment #1.

L107: What about the increasing trend at the shortest heights? Effects on extrapolation in the management scenarios?

→ In the specific case mentioned by the reviewer, we found some sort of compensation between height and biomass. The two variables sampled over the whole gradient show a strong positive correlation (Spearman rank's correlation = 0.70, p-value<0.05) and therefore increasing heights generally lead to increasing biomass loss, as expected. However, at the shortest heights (height<16 meters), we found a slight opposite relation (Spearman rank correlation = -0.16, p-value=0.11) (Fig. R3, upper panel). These occurrences in the observed disturbance database are associated to older trees (Fig. R3, lower panel) that typically have large diameters, which ultimately lead – despite the limited tree height - to relatively high biomass stands (and then to relatively high BL_{rel}). All these interaction terms cannot be visualized in 1-dimensional PDPs as above-mentioned, but they are explicitly included in our RF models (see our response to your comments #3 and #5).

Figure R3. Variability of biomass values and tree age across a gradient of height as recorded in the FORWIND database. The tops and bottoms of each “box” are the 25th and 75th percentiles of the samples, respectively. The line in the middle of each box is the sample median. Whiskers are drawn from the ends of the interquartile ranges to the furthest observations within the whisker length. Observations beyond the whisker length are marked as outliers here defined as a point that is more than 1.5 times the interquartile range away from the top or bottom of the box.

L115: The most important variable in trends of score, but MAD span 0 the entire range.

- Figures show zero-centered PDPs, therefore MAD span 0 by definition. Please, see our response to your major comment #1.

L115: Counter intuitive. Is this an artifact of the combination of insect attack on evergreen needleleaf trees but having broadleaf deciduous trees with high LAI and low beetle attack in the same sample?

Bark beetle attack generally increases with increasing tree density, provided the trees are large enough to sustain a beetle population, due to tree stress from greater competition for water, greater pheromone concentration leading to better beetle aggregation, and easier emerging beetle spread from tree to tree.

*Raffa, K. F., B. H. Aukema, B. J. Bentz, A. L. Carroll, J. A. Hicke, M. G. Turner, and W. H. Romme. 2008. Cross-scale drivers of natural disturbances prone to anthropogenic amplification: The dynamics of bark beetle eruptions. *BioScience* 58:501-517.*

*Boone, C. K., B. H. Aukema, J. Bohlmann, A. L. Carroll, and K. F. Raffa. 2011. Efficacy of tree defense physiology varies with bark beetle population density: a basis for positive feedback in eruptive species. *Canadian Journal of Forest Research* 41:1174-1188.*

- Response functions computed for each specific PFT do not show substantial differences, as most of the binned records used to train/validate our RF models are common to all PFTs. Therefore, based on our results, we cannot explain the emerging patterns because of artefacts due to samples of records with mixed PFTs. We hypothesize that the emerging patterns may be associated to the good health conditions and limited water stress, which are typically associated to forests with high LAI and tree density. We have corrected the text to highlight the apparent contradiction suggested by the reviewer (see lines 136-138).

L119: This reference does not contain results that link high LAI to good health conditions or reduced water stress.

- We noted that in the version initially submitted the reference number was wrongly linked to the article. We have now corrected the link.

L123: But there are no interaction terms. How is the combined effect assessed?

- Please, see our response to your major comment #3.

L126: The PDP does not show this. It shows cold regions having less loss.

- Please consider that PDPs are zero-centered, therefore forest in cold climate show higher vulnerability compared to those in warm climate.

L132: Where is the RF evaluation? Table of classification accuracy? Maps of differences between observed and predicted biomass loss? Plots of residuals vs predictors? How might the uncertainty or lack of fit in the RF models be affecting the predicted vulnerabilities?

The RF models only explain less than half the variability in biomass loss, what do you think accounts for the other half? How might that be affecting the predicted vulnerabilities?

It's very difficult to assess the rest of the manuscript without this information.

→ Please, see our response to your major comments #2 and #4.

L166: How is elevation associated with drought, which is shown to be an important driver in the reference cited?

→ Based on the records of the IDS-USDA database we found that disturbances occurred at low and high elevation sites are also those which have experienced higher drought conditions, quantified both in terms of anomalies in average temperature and average SPEI (see Figure R4 below). Please see also our response your comment #5 regarding the interactions between multiple drivers.

Figure R4. Variability of average temperature anomalies and SPEI across a gradient of elevation as recorded in the IDS-USDA records. The tops and bottoms of each “box” are the 25th and 75th percentiles of the samples, respectively. The line in the middle of each box is the sample median. Whiskers are drawn from the ends of the interquartile ranges to the furthest observations within the whisker length. Observations beyond the whisker length are marked as outliers here defined as a point that is more than 1.5 times the interquartile range away from the top or bottom of the box.

L168-171: This statement, and the rest of the paper, is hard to assess without addressing the points in the section on response functions.

→ Please, see our responses to your major comments, in particular #1.

L172: The methods do not allow for novel conditions. For example, if the existing old trees are also very dense, there are no data to train on where old trees are less dense. This limits the interpretability of the scenario results and should be acknowledged.

→ After some concern raised by rev. 1 regarding the confusion of our vulnerability estimates with risk levels, we decided to remove this part. Please, see our response to your major comment #6.

L188: Interaction terms in the RF between tree density and tree age would help elucidate this possibility.

→ This text passage has been removed. Please see our response to your comment #3 concerning the interaction in RF models.

L510: But they are not independent. windthrow often leads to insect outbreak.

- We agree with the reviewer on the potential relevance of multiple interacting disturbances. Disturbance agents are rarely independent but tend to exacerbate each other in space and time. A strong interaction effect has been documented for instance between windthrows and bark beetle disturbances. Uprooted trees are virtually defenseless against bark beetles and provide readily available breeding material that promotes the build-up of beetle populations and the consequent increase in vulnerability to insect outbreaks^{5,6}. Insect outbreaks, in turn, may potentially affect the severity of subsequent forest fires by altering the abundance of potential available fuel⁷. The magnitude of these effects varies with insect type and timing. Therefore, as changes in the climate system are conducive to an increase in disturbance regimes, an intensification of compound effects originating from multiple disturbance interactions can be expected as well⁸⁻¹⁰. Cascading and amplification effects originating from natural disturbance interactions have the potential to exacerbate the vulnerability of forest ecosystems and lead to irreversible shifts in ecosystem states¹¹. However, to account properly for such interacting processes, large datasets of concurrent (or linked) disturbances are required, which, unfortunately, are currently missing.
- Given the current lack of such systematic monitoring systems of forest disturbances for Europe, in this study the unique “known” interactions explicitly considered are the effects of droughts on wildfires and insect outbreaks. However, droughts are treated in this study as a climate stressor and not a specific disturbance causing biomass loss. Therefore, our results only partially capture the effects of interacting disturbances and should be viewed in light of these limitations. We have clarified this in the text (lines 289-294 and lines 615-625).
- Recent research initiatives launched by the European Commission aimed at mapping natural disturbances in European Forests (e.g., Database of European Forest Insect & Disease Disturbances (DEFID2), <https://forest.jrc.ec.europa.eu/en/>) will pave the road for a quantitative assessment of such interacting processes.

Figures

Fig. 1: This is a very nice figure. It present a lot of information very clearly. I suggest adding the MAD shading on the response functions and a horizontal line at 0 to more easily see where that shading spans 0. As noted on Supplemental Figure 2, I believe the y-axis is the log-odds of being in a positive biomass lass category. Labeling it as percent loss is confusing.

- ➔ Thank you for your comment! We implemented your suggestion to add shading to the MAD response function but it ended up being more difficult to read and we had to refrain from using your suggestion. In order to clarify that response functions refer to zero-centered PDPs we instead included offset values in the panels.
- ➔ The variable and unit of the y-axis are currently represented. Our work deal with a random forest regression application, not with a classification task (e.g., see line 70, line 445, line 477, line 494). The response variable is a continuous metric and not a probability class. We have also added a new section in Methods (“Reconstruction of annual biomass time series”) to better describe the approach used to derive the response variable. We believe that this new material, added in response to a specific comment from another reviewer, may also contribute to better understand the type of variable shown over the y-axis.

Fig. 3: Another nice figure. Why not show the full distribution here?

- ➔ We have removed this figure and the corresponding text. Please, see our response to your comment #6.

Fig. S1: This is a helpful figure.

- ➔ Thank you, we agree, especially to simplify reproducibility of our analyses.

Fig. S2-S4: The y-axis is not in percent loss. It is, I think, the log-odds of belonging to a positive loss class. Add line at 0 to more easily see where MAD span 0.

- ➔ The variable and unit of the y-axis are correctly represented. Our work builds upon a random forest regression application, not a classification task. The response variable is a continuous metric and not a probability class. Our work deal with a random forest regression application, not with a classification task (e.g., see line 70, line 445, line 477, line 494). We have also added a new section in Methods to better describe the approach used to derive the response variable. We believe that this new material, added in response to a specific comment form another review, may also contribute to better understand the type of variable shown over the y-axis.
- ➔ Please, see also our response to your major comment #1.

Side remark: please note that uncertainty bounds in figure 5 have been corrected.

References

1. IPCC. Managing the Risks of Extreme Events and Disasters to Advance Climate Change Adaptation: Special Report of the Intergovernmental Panel on Climate Change. (Cambridge University Press, 2012).
2. Stadelmann, G., Bugmann, H., Wermelinger, B., Meier, F. & Bigler, C. A predictive framework to assess spatio-temporal variability of infestations by the European spruce bark beetle. *Ecography* **36**, 1208–1217 (2013).
3. Seidl, R. et al. Small beetle, large-scale drivers: how regional and landscape factors affect outbreaks of the European spruce bark beetle. *Journal of Applied Ecology* **53**, 530–540 (2016).
4. Biedermann, P. H. W. et al. Bark Beetle Population Dynamics in the Anthropocene: Challenges and Solutions. *Trends in Ecology & Evolution* **34**, 914–924 (2019).
5. Stadelmann, G., Bugmann, H., Wermelinger, B. & Bigler, C. Spatial interactions between storm damage and subsequent infestations by the European spruce bark beetle. *Forest Ecology and Management* **318**, 167–174 (2014).
6. Seidl, R. et al. Forest disturbances under climate change. *Nature Clim. Change* **7**, 395–402 (2017).
7. Meigs, G. W., Zald, H. S. J., Campbell, J. L., Keeton, W. S. & Kennedy, R. E. Do insect outbreaks reduce the severity of subsequent forest fires? *Environ. Res. Lett.* **11**, 045008 (2016).
8. Seidl, R., Schelhaas, M.-J., Rammer, W. & Verkerk, P. J. Increasing forest disturbances in Europe and their impact on carbon storage. *Nature Clim. Change* **4**, 806–810 (2014).
9. Westerling, A. L., Turner, M. G., Smithwick, E. A. H., Romme, W. H. & Ryan, M. G. Continued warming could transform Greater Yellowstone fire regimes by mid-21st century. *PNAS* **108**, 13165–13170 (2011).
10. Zscheischler, J. et al. Future climate risk from compound events. *Nature Climate Change* **8**, 469 (2018).
11. Seidl, R., Spies, T. A., Peterson, D. L., Stephens, S. L. & Hicke, J. A. REVIEW: Searching for resilience: addressing the impacts of changing disturbance regimes on forest ecosystem services. *Journal of Applied Ecology* **53**, 120–129 (2016).
12. EUROPE, F. State of Europe's Forests 2015 Report. (2015).
13. Schelhaas, M.-J., Nabuurs, G.-J. & Schuck, A. Natural disturbances in the European forests in the 19th and 20th centuries. *Global Change Biology* **9**, 1620–1633 (2003).
14. Peltola, H., Kellomäki, S., Hassinen, A. & Granander, M. Mechanical stability of Scots pine, Norway spruce and birch: an analysis of tree-pulling experiments in Finland. *Forest Ecology and Management* **135**, 143–153 (2000).
15. Fernandes, P. M. Combining forest structure data and fuel modelling to classify fire hazard in Portugal. *Annals of Forest Science* **66**, 415–415 (2009).
16. Walker, X. J. et al. Fuel availability not fire weather controls boreal wildfire severity and carbon emissions. *Nature Climate Change* 1–7 (2020) doi:10.1038/s41558-020-00920-8.
17. Goldstein, A., Kapelner, A., Bleich, J. & Pitkin, E. Peeking Inside the Black Box: Visualizing Statistical Learning With Plots of Individual Conditional Expectation. *Journal of Computational and Graphical Statistics* **24**, 44–65 (2015).
18. Breiman, L. Random Forests. *Machine Learning* **45**, 5–32 (2001).
19. Hastie, T., Tibshirani, R. & Friedman, J. *The Elements of Statistical Learning: Data Mining, Inference, and Prediction*, Second Edition. (Springer-Verlag, 2009).
20. Friedman, J. H. & Popescu, B. E. Predictive Learning via Rule Ensembles. *The Annals of Applied Statistics* **2**, 916–954 (2008).

Reviewer comments, second round:

Reviewer #3 (Remarks to the Author):

Dear Authors,

I appreciate the work you have done to address my concerns. I especially appreciate the more detailed explanation of your ML methods. Including this more detailed description in the text will allow readers who are less familiar with this modeling technique to better understand the conclusions and limitations of your research. I have no further concerns.

As an aside, I hope improved techniques for diagnosing ML models are soon available. It's important to be able to diagnose if the models are getting the right answers for the right reasons.